# An Infinite-Feature Extension for Bayesian ReLU Nets That Fixes Their Asymptotic Overconfidence

**Agustinus Kristiadi**
University of Tübingen
agustinus.kristiadi@uni-tuebingen.de

**Matthias Hein**
University of Tübingen
matthias.hein@uni-tuebingen.de

**Philipp Hennig**
University of Tübingen and MPI for Intelligent Systems, Tübingen
philipp.hennig@uni-tuebingen.de

## Abstract

A Bayesian treatment can mitigate overconfidence in ReLU nets around the training data. But far away from them, ReLU Bayesian neural networks (BNNs) can still underestimate uncertainty and thus be asymptotically overconfident. This issue arises since the output variance of a BNN with finitely many features is quadratic in the distance from the data region. Meanwhile, Bayesian linear models with ReLU features converge, in the infinite-width limit, to a particular Gaussian process (GP) with a variance that grows cubically so that no asymptotic overconfidence can occur. While this may seem of mostly theoretical interest, in this work, we show that it can be used in practice to the benefit of BNNs. We extend finite ReLU BNNs with infinite ReLU features via the GP and show that the resulting model is asymptotically maximally uncertain far away from the data while the BNNs' predictive power is unaffected near the data. Although the resulting model approximates a full GP posterior, thanks to its structure, it can be applied *post-hoc* to any pre-trained ReLU BNN at a low cost.

## 1 Introduction

Approximate Bayesian methods, which turn neural networks (NNs) into Bayesian neural networks (BNNs), can be used to address the overconfidence issue of NNs [1]. Specifically, Kristiadi et al. [2] recently showed for binary ReLU classification networks that far away from the training data, i.e. when scaling any input with a scalar $\alpha > 0$ and taking the limit $\alpha \to \infty$, the confidence of (Gaussian-based) BNNs is strictly less than one—"being Bayesian" can thus mitigate overconfidence. This result is encouraging vis-à-vis standard point-estimated networks, for which Hein et al. [3] showed earlier that the same asymptotic limit always yields arbitrarily high confidence. Nevertheless, BNNs can still be asymptotically overconfident, albeit less so than standard NNs, since the aforementioned uncertainty bound can be loose.

We identify that this issue arises because the variance over function outputs of a BNN is asymptotically quadratic, while the corresponding mean is asymptotically linear w.r.t. $\alpha$. Intuitively, fixing this issue requires adding an unbounded number of ReLU features with increasing distance from the training data, so that the output mean stays unchanged but the associated variance grows super-quadratically. And indeed there is a particular Gaussian process (GP), arising from the cubic spline kernel [4], which has cubic variance growth and can be seen as a Bayesian linear model with countably infinite ReLU features. In the context of the analysis, the fact that standard ReLU BNNs only use *finitely* many features makes them "miss out" on some uncertainty that should be there. In this work, we "add back" this missing uncertainty into finite ReLU BNNs by first extending the cubic spline kernel

35th Conference on Neural Information Processing Systems (NeurIPS 2021).

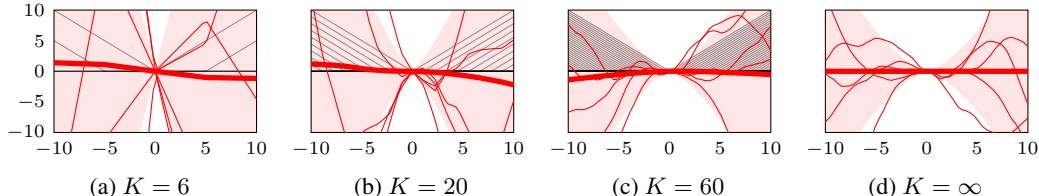

| (a) $K = 6$ | (b) $K = 20$ | (c) $K = 60$ | (d) $K = \infty$ |

Figure 1: The construction of a GP prior with the proposed "ReLU kernel", as the limiting covariance of the output of a Bayesian linear model with $K$ ReLU features (grey), arranged at regular intervals, oriented away from the origin. Red curves are function samples with the thick one being the mean, and the red shade their std. dev. With finite $K$ **(a-c)**, the variance grows quadratically, leading to the asymptotic overconfidence in ReLU BNNs. But, with $K = \infty$ **(d)**, the variance grows *cubically* away from the origin. The fact that this kernel has zero mean and negligible variance near the origin enables us to easily combine this GP with standard finite pre-trained ReLU BNNs.

to cover the whole input domain (Fig. 1) and then using the resulting GP to model residuals of BNNs [5–7]. Conceptually, we extend finite BNNs into infinite ones. The proposed kernel has two crucial properties: (i) It has negligible values around the origin, which we can assume without loss of generality to be the region where the data reside, and (ii) like the cubic spline kernel, its variance grows cubically in $\alpha$. Using the first property, we can approximately decompose the resulting *a posteriori* function output simply as *a posteriori* BNNs' output plus *a priori* the GP's output. This extension can therefore be applied to any pre-trained ReLU BNN in a *post-hoc* manner. And due to the second property, we can show that the extended BNNs are guaranteed to predict with uniform confidence away from the data. This approach thus fixes ReLU BNNs' asymptotic overconfidence, without affecting the BNNs' predictive mean. Finally, the method can be extended further while still preserving all these properties, by also modeling the representations of input points with the proposed GP. By doing so, the GP can adapt to the data well, and hence also improve the extended ReLU BNNs' non-asymptotic uncertainty.

A core contribution of this paper is the theoretical analysis: We show that our method (i) models the uncertainty that ReLU BNNs lack, thus (ii) ensuring that the surrounding output variance asymptotically grows cubically in the distance to the training data, and ultimately (iii) yields uniform asymptotic confidence in the multi-class classification setting. These results extend the prior analysis in so far as it is limited to the binary classification case and does not guarantee the asymptotically maximum-entropy prediction. Furthermore, our approach is complementary to the method of Meinke and Hein [8], which attains maximum uncertainty far from the data for non-Bayesian NNs. We empirically confirm the analysis and show effectiveness in the *non*-asymptotic regime.

## 2   Background

**Notation**   We denote a test point and its unknown label as $\boldsymbol{x}_*$ and $y_*$, respectively. We denote any quantity that depends on $\boldsymbol{x}_*$ with the same subscript, in particular $f_* := f(\boldsymbol{x}_*)$, $k_* := k(\boldsymbol{x}_*, \boldsymbol{x}_*)$.

### 2.1   Bayesian Neural Networks

We focus on multi-class classification problems. Let $f : \mathbb{R}^N \times \mathbb{R}^D \to \mathbb{R}^C$ defined by $(\boldsymbol{x}, \boldsymbol{\theta}) \mapsto f(\boldsymbol{x}; \boldsymbol{\theta}) =: f_{\boldsymbol{\theta}}(\boldsymbol{x})$ be a $C$-class ReLU network—a fully-connected or convolutional feed-forward network equipped with the ReLU nonlinearity. Here, $\boldsymbol{\theta}$ is the collection of all parameters of $f$. Given an i.i.d. dataset $\mathcal{D} := (\boldsymbol{x}_m, y_m)_{m=1}^{M}$, the standard training procedure amounts to finding a ***maximum a posteriori (MAP) estimate*** $\boldsymbol{\theta}_{\text{MAP}} = \arg\max_{\boldsymbol{\theta}} \log p(\boldsymbol{\theta} \mid \mathcal{D})$.

One can also apply Bayes' theorem to infer the full posterior distribution of $\boldsymbol{\theta}$—the resulting network is called a ***Bayesian neural network (BNN)***. A common way to approximate the posterior $p(\boldsymbol{\theta} \mid \mathcal{D})$ of a BNN is by a Gaussian $q(\boldsymbol{\theta}) := \mathcal{N}(\boldsymbol{\mu}, \boldsymbol{\Sigma})$. Given this approximate posterior and a test point $\boldsymbol{x}_* \in \mathbb{R}^N$, the prediction is given by $p(y_* \mid \boldsymbol{x}_*, \mathcal{D}) = \int \text{softmax}(f_{\boldsymbol{\theta}}(\boldsymbol{x}_*)) \, q(\boldsymbol{\theta}) \, d\boldsymbol{\theta}$. The Gaussian approximate posterior and softmax likelihood are our assumptions throughout this paper.

One can obtain a useful two-step closed-form approximation of the previous integral as follows. First, we perform a ***network linearization*** on $f$ around $\boldsymbol{\mu}$ and obtain the following marginal over $f(\boldsymbol{x}_*)$:

$$p(f_* \mid \boldsymbol{x}_*, \mathcal{D}) \approx \mathcal{N}(f_{\boldsymbol{\mu}}(\boldsymbol{x}_*), \boldsymbol{J}_*^\top \boldsymbol{\Sigma} \boldsymbol{J}_*), \tag{1}$$

where $\boldsymbol{J}_*$ is the $D \times C$ Jacobian matrix of $f_{\boldsymbol{\theta}}(\boldsymbol{x}_*)$ w.r.t. $\boldsymbol{\theta}$ at $\boldsymbol{\mu}$. For brevity, let $\boldsymbol{m}_*$ and $\boldsymbol{V}_*$ be the above mean and covariance. To obtain the predictive distribution, we then apply the ***generalized probit approximation*** [9, 10]:

$$p(y_* = c \mid \boldsymbol{x}_*, \mathcal{D}) = \int \mathrm{softmax}(f_*)_c \, p(f_* \mid \boldsymbol{x}_*, \mathcal{D}) \, df_* \approx \frac{\exp(m_{*c} \, \kappa_{*c})}{\sum_{i=1}^C \exp(m_{*i} \, \kappa_{*i})}, \tag{2}$$

where for each $i = 1, \dots, C$, the real number $m_{*i}$ is the $i$-th component of the vector $\boldsymbol{m}_*$, and $\kappa_{*i} := (1 + \pi/8 \, v_{*ii})^{-1/2}$ where $v_{*ii}$ is the $i$-th diagonal term of the matrix $\boldsymbol{V}_*$. Both approximations above have been shown to be good both in terms of their errors and predictive performance [10–13].[1]

While analytically useful, these approximations can be expensive due to the computation of the Jacobian matrix $\boldsymbol{J}_*$. Thus, ***Monte Carlo (MC) integration*** is commonly used as an alternative, i.e. we approximate $p(y_* \mid \boldsymbol{x}_*, \mathcal{D}) \approx 1/S \sum_{s=1}^S p(y_* \mid f_{\boldsymbol{\theta}_s}(\boldsymbol{x}_*))$; $\boldsymbol{\theta}_s \sim q(\boldsymbol{\theta})$. Finally, given a classification predictive distribution $p(y_* \mid \boldsymbol{x}_*, \mathcal{D})$, we define the predictive ***confidence*** of $\boldsymbol{x}_*$ as the maximum probability $\max_{c \in \{1,\dots,C\}} p(y_* = c \mid \boldsymbol{x}_*, \mathcal{D})$ over class labels. Far from the data, ideally the model should produce the ***uniform confidence*** $p(y_* = c \mid \boldsymbol{x}_*, \mathcal{D}) = 1/C$ for all $c = 1, \dots, C$.

## 2.2 Asymptotic Overconfidence in BNNs

Given a fixed point estimate $\boldsymbol{\theta}_{\mathrm{MAP}}$, the ReLU network $f_{\boldsymbol{\theta}_{\mathrm{MAP}}}$ yields overconfident predictions, even for points far away from the training data [3]. That is, for almost any input $\boldsymbol{x}_* \in \mathbb{R}^N$, one can show that there exists a class $c \in \{1, \dots, C\}$ such that $\lim_{\alpha \to \infty} \mathrm{softmax}(f_{\boldsymbol{\theta}_{\mathrm{MAP}}}(\alpha \boldsymbol{x}_*))_c = 1$. Intuitively, this issue arises because the ReLU network yields a piecewise-affine function with finitely many linear regions (the domain of each affine function). Under this setup, by scaling $\boldsymbol{x}_*$ with $\alpha$, at some point one arrives at an "outer linear region" and in this region, the network is always affine—either increasing or decreasing—even as $\alpha$ tends to infinity, and thus its softmax output converges to a "one-hot vector".

BNNs, even with a simple Gaussian approximate posterior, can help to mitigate this problem in binary classifications, as shown by Kristiadi et al. [2]. The crux of their proof is the observation that in an outer linear region, the predictive distribution (via the probit approximation) is given by[2]

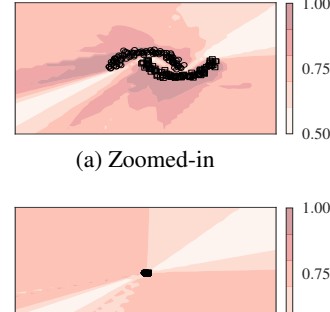

1.00
0.75
0.50

(a) Zoomed-in

1.00
0.75
0.50

(b) Zoomed-out

Figure 2: Confidence estimates of a BNN.

$$p(y_* = 1 \mid \alpha \boldsymbol{x}_*, \mathcal{D}) \approx \sigma \left( \frac{\alpha \boldsymbol{u}^\top \boldsymbol{x}_*}{\sqrt{1 + \pi/8 \, v(\alpha \boldsymbol{x}_*)}} \right), \tag{3}$$

where $\sigma$ is the logistic-sigmoid function, $\boldsymbol{u}$ is the parameter vector corresponding to the linear region and the quadratic function $v$ maps $\alpha \boldsymbol{x}_*$ to the variance of the network output. Unfortunately, both the numerator and denominator above are linear in $\alpha$ and thus altogether $p(y_* = 1 \mid \alpha \boldsymbol{x}_*, \mathcal{D})$ only converges to a constant strictly less than 1 as $\alpha \to \infty$, not necessarily the ideal uniform confidence prediction. BNNs can therefore still be overconfident, albeit less so than the point-estimated counterpart (Fig. 2).

## 2.3 ReLU and Gaussian processes

The ReLU activation function $\mathrm{ReLU}(z) := \max(0, z)$ [14] has become the *de facto* choice of non-linearity in deep learning. Given an arbitrary real number $c$, it can be generalized as $\mathrm{ReLU}(z; c) :=$

---

[1]The network linearization comes with the error of $O(\|\theta - \theta_{\mathrm{MAP}}\|^2)$ by Taylor's theorem. Meanwhile for the (generalized) probit approximation, low empirical error has been observed by [10, Fig. 1].

[2]We omit the bias parameter for simplicity.

$\max(0, z - c)$, with the "kink" at location $c$. An alternative formulation, useful below, is in terms of the Heaviside function $H$ as $\text{ReLU}(z; c) = H(z - c) \cdot (z - c)$. We may define a collection of $K$ such ReLU functions evaluated at some point in $\mathbb{R}$ as the function $\boldsymbol{\phi} : \mathbb{R} \to \mathbb{R}^K$ with $z \mapsto (\text{ReLU}(z; c_1), \dots, \text{ReLU}(z; c_K))^\top$. We call this function the **ReLU feature map**, which can be interpreted as "placing" ReLU functions at different locations in $\mathbb{R}$.

Consider a linear model $g : \mathbb{R} \times \mathbb{R}^K \to \mathbb{R}$ defined by $g(x; \boldsymbol{w}) := \boldsymbol{w}^\top \boldsymbol{\phi}(x)$. Suppose $\boldsymbol{\phi}$ regularly places $K$ generalized ReLU functions centered at $(c_i)_{i=1}^K$ on $[c_{\min}, c_{\max}] \subset \mathbb{R}$, where $c_{\min} < c_{\max}$. If we consider a Gaussian prior $p(\boldsymbol{w}) := \mathcal{N}\left(\boldsymbol{w} \,\middle|\, \boldsymbol{0}, \sigma^2 K^{-1}(c_{\max} - c_{\min})\boldsymbol{I}\right)$, then as $K \to \infty$, the distribution over $g$ is a Gaussian process with mean 0 and covariance (full derivation in Appendix A):

$$\widehat{k}^1(x, x'; c_{\min}, \sigma^2) := \sigma^2 H(\bar{x} - c_{\min}) \left( \frac{1}{3}(\bar{x}^3 - c_{\min}^3) - \frac{1}{2}(\bar{x}^2 - c_{\min}^2)(x + x') + (\bar{x} - c_{\min})xx' \right).$$

Here, the superscript 1 denotes the fact that this function is over a 1-dimensional input space and $\bar{x} := \min(x, x')$. Since the expression above does not depend on $c_{\max}$, we can consider the limit $c_{\max} \to \infty$, and thus this kernel is non-zero on $(c_{\min}, \infty)$. This covariance function is the **cubic spline kernel** [4]. The name indicates that posterior mean of the associated GP is piecewise-cubic. But it also has variance $\widehat{k}^1(x, x; c_{\min}, \sigma^2)$ which is cubic in $x$ and negligible for $x$ close to $c_{\min}$.

## 3 Infinite-Feature Extension for ReLU BNNs

From Section 2.2 it becomes clear that the asymptotic miscalibration of ReLU BNNs is due to the finite number of ReLU features used, which results in only quadratic variance growth. An infinite-ReLU GP with the cubic spline kernel has cubic variance growth, which, combined with the probit approximation, yields uniform confidence in the limit. But of course, full GP inference is prohibitively expensive. In this section, we propose a cheap, *post-hoc* way to extend any pre-trained ReLU BNN with the aforementioned GP by extending the cubic spline kernel and exploiting its two important properties. We will see that the resulting model approximates the full GP posterior and combines the predictive power of the BNN with a guarantee for asymptotically uniform confidence. While in our analysis we employ network linearization for analytical tractability, the method can be applied via MC-integration as well (cf. Section 5). All proofs are in Appendix B.

### 3.1 The Double-Sided Cubic Spline Kernel

The cubic spline kernel is one-sided in the sense that it has zero variance on $(-\infty, c_{\min})$, and therefore is unsuitable for modeling over the entire domain. This is easy to fix by first setting $c_{\min} = 0$ to obtain a kernel $\overrightarrow{k}^1(x, x'; \sigma^2) := \widehat{k}^1(x, x'; 0, \sigma^2)$ which is non-zero only on $(0, \infty)$. Now, by an entirely analogous construction with infinitely many ReLU functions pointing to the opposite direction (i.e. left) via $\text{ReLU}(-z; c)$, we obtain another kernel $\overleftarrow{k}^1(x, x'; \sigma^2) := \overrightarrow{k}^1(-x, -x'; \sigma^2)$, which is non-zero only on $(-\infty, 0)$. Combining them together, we obtain the following kernel, which covers the whole real line: $k^1(x, x'; \sigma^2) := \overleftarrow{k}^1(x, x'; \sigma^2) + \overrightarrow{k}^1(x, x'; \sigma^2)$—see Fig. 1. Note in particular that the variance $k^1(0, 0)$ at the origin is zero. This is a key feature of this kernel that enables us to efficiently combine the resulting GP prior with a pre-trained BNN.

For multivariate input domains, we define

$$k(\boldsymbol{x}, \boldsymbol{x}'; \sigma^2) := \frac{1}{N} \sum_{i=1}^N k^1(x_i, x_i'; \sigma^2) \tag{4}$$

for any $\boldsymbol{x}, \boldsymbol{x}' \in \mathbb{R}^N$ with $N > 1$. We here deliberately use a summation, instead of the alternative of a product, since we want the associated GP to add uncertainty whenever *at least* one input dimension has non-zero value. (By contrast, a product $k(\boldsymbol{x}, \boldsymbol{x}')$ is zero if one of the $k^1(x_i, x_i')$ is zero.) We call this kernel the **double-sided cubic spline (DSCS) kernel**. Similar to the one-dimensional case, two crucial properties of this kernel are that it has negligible variance around the origin of $\mathbb{R}^N$ and for any $\boldsymbol{x}_* \in \mathbb{R}^N$ and $\alpha \in \mathbb{R}$, the value $k(\alpha\boldsymbol{x}_*, \alpha\boldsymbol{x}_*)$ is *cubic* in $\alpha$.

## 3.2 ReLU-GP Residual

For simplicity, we start with real-valued BNNs and discuss the generalization to multi-dimensional output later. Let $f : \mathbb{R}^N \times \mathbb{R}^D \to \mathbb{R}$ be an $L$-layer, real-valued ReLU BNN. Since $f$ by itself can be asymptotically overconfident, it has *residual* in its uncertainty estimates far from the data. Our goal is to extend $f$ with the GP prior that arises from the DSCS kernel, to model this uncertainty residual. We do so by placing infinitely many ReLU features over its input space $\mathbb{R}^N$ by following the DSCS kernel construction in the previous section. Then, we arrive at a zero-mean GP prior $\mathcal{GP}(\widehat{f} \mid 0, k)$ over a real-valued random function $\widehat{f} : \mathbb{R}^N \to \mathbb{R}$. Following previous works [15, 6, 7], we use this GP prior to model the residual of $f$ by defining

$$\widetilde{f} := f + \widehat{f}, \qquad \text{where } \widehat{f} \sim \mathcal{GP}(0, k), \tag{5}$$

and call this method ***ReLU-GP residual (RGPR)***.

We now analyze RGPR. Besides linearization, we assume that the DSCS kernel has, without loss of generality, a negligibly small value at the data, i.e. $k(\boldsymbol{x}_m, \boldsymbol{x}_*) \approx 0$ for all $(\boldsymbol{x}_m)_{m=1}^M$ and any i.i.d. test point $\boldsymbol{x}_*$. Note that this can always be satisfied by centering and scaling. The error of this approximation is stated in the following.

**Lemma 1.** *Let $0 < \delta < 1$, and let $\sigma^2 > 0$ be a constant. For any $\boldsymbol{x}, \boldsymbol{x}' \in \mathbb{R}^N$ with $\|\boldsymbol{x}\|^2, \|\boldsymbol{x}'\|^2 \le \delta$ we have $k(\boldsymbol{x}, \boldsymbol{x}'; \sigma^2) \in O(\delta^3)$.*

Using this approximation, we show the approximate GP posterior of $\widetilde{f}$.

**Proposition 2 (RGPR's GP Posterior).** *Let $f : \mathbb{R}^N \times \mathbb{R}^D \to \mathbb{R}$ be a ReLU BNN with weight distribution $\mathcal{N}(\boldsymbol{\theta} \mid \boldsymbol{\mu}, \boldsymbol{\Sigma})$, and let $\mathcal{D} := (\boldsymbol{x}_m, y_m)_{m=1}^M =: (\boldsymbol{X}, \boldsymbol{y})$ be a dataset. Assume that $\|\boldsymbol{x}_m\|^2, \|\boldsymbol{x}\|^2 \le \delta$ for all $m = 1, \dots, M$ and any i.i.d. test point $\boldsymbol{x} \in \mathbb{R}^N$, with $0 < \delta < 1$. Then given an i.i.d. input point $\boldsymbol{x}_* \in \mathbb{R}^N$, under the linearization of $f$ w.r.t. $\boldsymbol{\theta}$ around $\boldsymbol{\mu}$, the GP posterior over $\widetilde{f}_*$ is a Gaussian with mean and variance*

$$\mathbb{E}(\widetilde{f}_* \mid \mathcal{D}) \approx f(\boldsymbol{x}_*; \boldsymbol{\mu}) + \boldsymbol{h}_*^\top \boldsymbol{C}^{-1}(\boldsymbol{y} - f(\boldsymbol{X}; \boldsymbol{\mu})), \tag{6}$$

$$\mathrm{Var}(\widetilde{f}_* \mid \mathcal{D}) \approx \boldsymbol{g}(\boldsymbol{x}_*)^\top \boldsymbol{\Sigma} \boldsymbol{g}(\boldsymbol{x}_*) + k(\boldsymbol{x}_*, \boldsymbol{x}_*) - \boldsymbol{h}_*^\top \boldsymbol{C}^{-1} \boldsymbol{h}_*, \tag{7}$$

*respectively, where $\boldsymbol{h}_* := (\mathrm{Cov}(f(\boldsymbol{x}_*), f(\boldsymbol{x}_1)), \dots, \mathrm{Cov}(f(\boldsymbol{x}_*), f(\boldsymbol{x}_M)))^\top$, while $\boldsymbol{C}$ is the covariance matrix $(\mathrm{Cov}(f(\boldsymbol{x}_i), f(\boldsymbol{x}_j)))_{ij}^M$, and $f(\boldsymbol{X}; \boldsymbol{\mu}) := (f(\boldsymbol{x}_1; \boldsymbol{\mu}), \dots, f(\boldsymbol{x}_M; \boldsymbol{\mu}))^\top$. Moreover, the approximation error in (6) is in $O\left((\delta^6 \|\boldsymbol{C}^{-1}\|\|\boldsymbol{m}\|)/(1 - \delta^3 \|\boldsymbol{C}^{-1}\|)\right)$ where $\boldsymbol{m} = \boldsymbol{C}^{-1}(\boldsymbol{y} - f(\boldsymbol{X}; \boldsymbol{\mu}))$, while the error in (7) is in $O\left((\delta^6(\|\boldsymbol{C}^{-1}\| + \|\boldsymbol{C}^{-1}\|\|\boldsymbol{m}\|))/(1 - \delta^3 \|\boldsymbol{C}^{-1}\|)\right)$ where $\boldsymbol{m} = \boldsymbol{C}^{-1}\boldsymbol{h}_*$.*

While this result is applicable to any Gaussian weight distribution, an interesting special case is where we assume that the BNN is *well-trained*, i.e. we have a Gaussian (approximate) posterior $p(\boldsymbol{\theta} \mid \mathcal{D})$ which induces accurate prediction and high output confidence on each of the training data. In this case, the last term of (6) is negligible since the residual $\boldsymbol{y} - f(\boldsymbol{X}; \boldsymbol{\mu})$ is close zero. Moreover, notice that the last term in (7) can be upper-bounded by

$$\boldsymbol{h}_*^\top \boldsymbol{C}^{-1} \boldsymbol{h}_* \le \lambda_{\max} \|\boldsymbol{h}_*\|^2 = \lambda_{\max} \sum_{m=1}^M \mathrm{Cov}(f(\boldsymbol{x}_*), f(\boldsymbol{x}_m))^2,$$

where $\lambda_{\max}$ denotes the largest eigenvalue of $\boldsymbol{C}^{-1}$. The last summand above can further be upper-bounded via the Cauchy-Schwarz inequality by $\mathrm{Cov}(f(\boldsymbol{x}_*), f(\boldsymbol{x}_m))^2 \le \mathrm{Var}(f(\boldsymbol{x}_*))\mathrm{Var}(f(\boldsymbol{x}_m))$. But our assumption implies that $\mathrm{Var}(f(\boldsymbol{x}_m))$ is close to zero for all $m = 1, \dots, M$. Thus, if $f$ is a pre-trained ReLU BNN, we approximately have

$$\widetilde{f}_* \sim \mathcal{N}(f(\boldsymbol{x}_*; \boldsymbol{\mu}), \boldsymbol{g}_*^\top \boldsymbol{\Sigma} \boldsymbol{g}_* + k_*), \tag{8}$$

which can be thought of as arising from the sum of two Gaussian r.v.s. $f_* \sim \mathcal{N}(f(\boldsymbol{x}_*; \boldsymbol{\mu}), \boldsymbol{g}_*^\top \boldsymbol{\Sigma} \boldsymbol{g}_*)$ and $\widehat{f}_* \sim \mathcal{N}(0, k_*)$—we are back to the definition of RGPR (5). Thus, unlike previous works on modeling residuals with GPs [15, 6, 7], the *GP posterior* of RGPR can approximately be written as *a posteriori* $f$ plus *a priori* $\widehat{f}$. RGPR can hence be applied *post-hoc*, after the usual training process of

the BNN. Furthermore, we see that RGPR does indeed model only the uncertainty residual of the BNN since it only affects the predictive variance. In particular, it does not affect the output mean of the BNN and thus preserves its predictive accuracy—this is often desirable in practice since the main reason for using deep ReLU nets is due to their accurate predictions.

Generalization to BNNs with multiple outputs is straightforward. Let $f : \mathbb{R}^N \times \mathbb{R}^D \to \mathbb{R}^C$ be a vector-valued, pre-trained, $L$-layer ReLU BNN with posterior $\mathcal{N}(\boldsymbol{\theta} \mid \boldsymbol{\mu}, \boldsymbol{\Sigma})$. We assume that the following real-valued random functions $(\widehat{f}^{(c)} : \mathbb{R}^{\widehat{N}} \to \mathbb{R})_{c=1}^C$ are i.i.d. as the GP prior $\mathcal{GP}(0, k)$ (5). Thus, for any $\boldsymbol{x}_* \in \mathbb{R}^N$, defining $\widehat{f}_* := (\widehat{f}_*^{(1)}, \ldots, \widehat{f}_*^{(C)})^\top$, we have $p(\widehat{f}_*) = \mathcal{N}(\boldsymbol{0}, k_* \boldsymbol{I})$, and so under the linearization of $f$, this implies that the marginal GP posterior of RGPR is approximately given by the following $C$-variate Gaussian

$$p(\widetilde{f}_* \mid \boldsymbol{x}_*, \mathcal{D}) \approx \mathcal{N}(f_{\boldsymbol{\mu}}(\boldsymbol{x}_*), \boldsymbol{J}_*^\top \boldsymbol{\Sigma} \boldsymbol{J}_* + k_* \boldsymbol{I}). \tag{9}$$

We can do so since intuitively (9) is simply obtained as a result of "stacking" $C$ independent $\widetilde{f}_*^{(c)}$'s, each of which satisfies Proposition 2. The following lemma shows that asymptotically, the marginal variances of $\widetilde{f}_*$ grow cubically as we scale the test point.

**Lemma 3 (Asymptotic Variance Growth).** *Let $f : \mathbb{R}^N \times \mathbb{R}^D \to \mathbb{R}^C$ be a pre-trained ReLU network with posterior $\mathcal{N}(\boldsymbol{\theta} \mid \boldsymbol{\mu}, \boldsymbol{\Sigma})$ and $\widetilde{f}$ be obtained from $f$ via RGPR. Suppose that the linearization of $f$ w.r.t. $\boldsymbol{\theta}$ around $\boldsymbol{\mu}$ is employed. For any $\boldsymbol{x}_* \in \mathbb{R}^N$ with $\boldsymbol{x}_* \neq \boldsymbol{0}$ there exists $\beta > 0$ such that for any $\alpha \geq \beta$ and each $c = 1, \ldots, C$, the variance $\mathrm{Var}(\widetilde{f}^{(c)}(\alpha \boldsymbol{x}_*))$ under (9) is in $\Theta(\alpha^3)$.*

Equipped with this result, we are now ready to state our main result. The following theorem shows that RGPR yields the ideal asymptotic uniform confidence of $1/C$ given any pre-trained ReLU classification BNN with an arbitrary number of classes.

**Theorem 4 (Uniform Asymptotic Confidence).** *Let $f : \mathbb{R}^N \times \mathbb{R}^D \to \mathbb{R}^C$ be a $C$-class pre-trained ReLU network equipped with the posterior $\mathcal{N}(\boldsymbol{\theta} \mid \boldsymbol{\mu}, \boldsymbol{\Sigma})$ and let $\widetilde{f}$ be obtained from $f$ via RGPR. Suppose that the linearization of $f$ and the generalized probit approximation (2) is used for approximating the predictive distribution $p(y_* = c \mid \alpha \boldsymbol{x}_*, \widetilde{f}, \mathcal{D})$ under $\widetilde{f}$. For any input $\boldsymbol{x}_* \in \mathbb{R}^N$ with $\boldsymbol{x}_* \neq \boldsymbol{0}$ and for every class $c = 1, \ldots, C$, we have $\lim_{\alpha \to \infty} p(y_* = c \mid \alpha \boldsymbol{x}_*, \widetilde{f}, \mathcal{D}) = 1/C$.*

As a sketch of the proof for this theorem, consider the special case of binary classification. Here, we notice that the variance $v$ in the probit approximation (3) is now a cubic function of $\alpha$ under RGPR, due to Lemma 3. Thus, it is easy to see that the term inside of $\sigma$ decays like $1/\sqrt{\alpha}$ far away from the training data. Therefore, in this case, $p(y = 1 \mid \alpha \boldsymbol{x}_*, \mathcal{D})$ evaluates to $\sigma(0) = 1/2$ as $\alpha \to \infty$, and hence we obtain the asymptotic maximum entropy prediction.

We remark that the pre-trained assumption on $f$ in Lemma 3 and Theorem 4 can be removed. Intuitively, this is because under the scaling of $\alpha$ on $\boldsymbol{x}_*$, the last term of (7) is in $\Theta(\alpha^2)$. Thus, it is asymptotically dominated by the $\Theta(\alpha^3)$ growth induced by the DSCS kernel in the second term. We however present the statements as they are since they support the *post-hoc* spirit of RGPR.

### 3.3 Extending RGPR to Non-Asymptotic Regimes

While the previous construction is sufficient for modeling uncertainty far away from the data, it does not necessarily model the uncertainty *near* the data region well. Figure 3(a) shows this behavior: the variance of the GP prior equipped with the DSCS kernel grows slowly around the data and hence, even though Theorem 4 will still apply in the limit, RGPR has a minimal effect on the uncertainty of the BNN in non-asymptotic regimes.

A way to address this is to adapt RGPR's notion of proximity between input points. This can be done by using the higher-level data representations already available from the pre-trained NN—a test point close to the data in the input space can be far from them in the representation space, thereby the DSCS kernel might assign a large variance. Based on this intuition, we extend RGPR by additionally

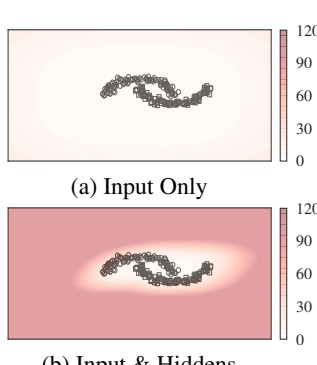

(a) Input Only

(b) Input & Hiddens

Figure 3: Variance of $\widehat{f}$.

placing infinite ReLU features on the representation spaces of the point-estimated network $f_{\boldsymbol{\mu}}$ induced by the BNN $f$, where $\boldsymbol{\mu}$ is the mean of the Gaussian posterior of $f$, as follows.

For each $l = 1, \ldots, L-1$ and any input $\boldsymbol{x}_*$, let $N_l$ be the size of the $l$-th hidden layer of $f_{\boldsymbol{\mu}}$ and $\boldsymbol{h}_*^{(l)}$ be the $l$-th hidden representation of $\boldsymbol{x}_*$. By convention, we assume that $N_0 := N$ and $\boldsymbol{h}_*^{(0)} := \boldsymbol{x}_*$. Now, we place for each $l = 0, \ldots, L-1$ an infinite number of ReLU features on the representation space $\mathbb{R}^{N_l}$, and thus we obtain a random function $\widehat{f}^{(l)} : \mathbb{R}^{N_l} \to \mathbb{R}$ distributed as $\mathcal{GP}(0, k)$. Then, given that $\widehat{N} := \sum_{l=0}^{L-1} N_l$, we define $\widehat{f} : \mathbb{R}^{\widehat{N}} \to \mathbb{R}$ by $\widehat{f} := \widehat{f}^{(0)} + \cdots + \widehat{f}^{(L-1)}$, i.e. we assume that $\{\widehat{f}^{(l)}\}_{l=0}^{L-1}$ are independent. This function is therefore a function over *all* representation (including the input) spaces of $f_{\boldsymbol{\mu}}$, distributed as the additive Gaussian process $\mathcal{GP}(0, \sum_{l=0}^{L-1} k)$. In other words, given all representations $\boldsymbol{h}_* := (\boldsymbol{h}_*^{(l)})_{l=0}^{L-1}$ of $\boldsymbol{x}_*$ under $f_{\boldsymbol{\mu}}$, the marginal over the function output $\widehat{f}(\boldsymbol{h}_*)$ is given by

$$p(\widehat{f}_*) = \mathcal{N}\left(0, \sum_{l=0}^{L-1} k\left(\boldsymbol{h}_*^{(l)}, \boldsymbol{h}_*^{(l)}; \sigma_l^2\right)\right). \tag{10}$$

We can then use this definition of $\widehat{f}$ as a drop-in replacement in (5) to define RGPR. Figure 3(b) visualizes the effect: the low-variance region modeled by $\widehat{f}$ becomes more compact around the data.

The analysis from the previous section still applies here since it is easy to see that the variance of $\widehat{f}_*$ in (10) is still cubic in $\alpha$. In practice, however, it is not necessarily true anymore that each $\boldsymbol{h}_*^{(l)}$ is close to the origin in $\mathbb{R}^{N_l}$. To fix this, one can center and scale each $\boldsymbol{h}_*^{(l)}$ via standardization using the mean $\mathbb{E}_{\boldsymbol{x} \in \mathcal{D}}(\boldsymbol{h}^{(l)}(\boldsymbol{x}))$ and scaled standard deviation $r\sqrt{\mathrm{Var}_{\boldsymbol{x} \in \mathcal{D}}(\boldsymbol{h}^{(l)}(\boldsymbol{x}))}$ with $r > 1$, before evaluating the kernel in (10) (these quantities only need to be computed once). Note that by tuning the DSCS kernel's hyperparameter $\sigma^2$ such that confidence over the training data is preserved (cf. the next section), RGPR becomes insensitive to the choice of $r$ since intuitively the tuning procedure will make sure that the DSCS kernel does not assign large variance to the training data. Therefore, in practice we set $r = 1$.

Algorithm 1 provides a pseudocode of RGPR for classification predictions via MC-integration. The only overhead compared to the usual MC-integrated BNN prediction step are (marked in red) (i) a single additional forward-pass over $f_{\boldsymbol{\mu}}$, (ii) $L$ evaluations of the DSCS kernel $k$, and (iii) sampling from a $C$-dimensional diagonal Gaussian. Their costs are negligible compared to the cost of obtaining the standard MC-prediction of $f$, which, in particular, requires multiple forward passes.

### 3.4 Hyperparameter Tuning

The kernel hyperparameters $(\sigma_l^2)_{l=0}^{L-1} =: \boldsymbol{\sigma}^2$ control the variance growth of the DSCS kernel. Since RGPR is a GP model, one way to tune $\boldsymbol{\sigma}^2$ is via marginal likelihood maximization. However, this leads to an expensive procedure even if a stochastic approximation [16] is employed since the computation of the RGPR kernel (9) requires the network's Jacobian and the explicit kernel matrix need to be formed. Note however that those quantities are not needed for the computation of the predictive distribution via MC-integration (Algorithm 1). Hence, a cheaper yet still valid option to tune $\boldsymbol{\sigma}^2$ is to use a cross-validation (CV) which depends only on predictions over validation data $\mathcal{D}_{\mathrm{val}}$ [17, Ch. 5].

A straightforward way to perform CV is by maximizing the validation log-likelihood (LL). That is, we maximize the objective $\mathcal{L}_{\mathrm{LL}}(\boldsymbol{\sigma}^2) :=$

---

**Algorithm 1** MC-prediction for RGPR. Differences from the standard procedure are in **red**.

**Input:**
    Pre-trained $L$-layer, ReLU BNN classifier $f$ with posterior $\mathcal{N}(\boldsymbol{\theta} \mid \boldsymbol{\mu}, \boldsymbol{\Sigma})$. Test point $\boldsymbol{x}_* \in \mathbb{R}^N$. Centering and scaling function std. Hyperparameters $(\sigma_l^2)_{l=0}^{L-1}$. Number of MC samples $S$.
1: $(\boldsymbol{h}_*^{(l)})_{l=1}^{L-1} = \texttt{forward}(f_{\boldsymbol{\mu}}, \boldsymbol{x}_*)$
2: $v_s(\boldsymbol{x}_*) = \sum_{l=0}^{L-1} k(\texttt{std}(\boldsymbol{h}_*^{(l)}), \texttt{std}(\boldsymbol{h}_*^{(l)}); \sigma_l^2)$
3: **for** $s = 1, \ldots, S$ **do**
4:     $\boldsymbol{\theta}_s \sim \mathcal{N}(\boldsymbol{\theta} \mid \boldsymbol{\mu}, \boldsymbol{\Sigma})$
5:     $\boldsymbol{f}_s(\boldsymbol{x}_*) = f(\boldsymbol{x}_*; \boldsymbol{\theta}_s)$
6:     $\widehat{f}_s(\boldsymbol{x}_*) \sim \mathcal{N}(\boldsymbol{0}, v_s(\boldsymbol{x}_*)\boldsymbol{I})$
7:     $\widetilde{f}_s(\boldsymbol{x}_*) = f_s(\boldsymbol{x}_*) + \widehat{f}_s(\boldsymbol{x}_*)$
8: **end for**
9: **return** $1/S \sum_{s=1}^{S} \mathrm{softmax}(\widetilde{f}_s(\boldsymbol{x}_*))$

---

$\sum_{\boldsymbol{x}_*, y_* \in \mathcal{D}_{\mathrm{val}}} \log p(y_* \mid \boldsymbol{x}_*, \mathcal{D}; \boldsymbol{\sigma}^2)$. However, this tends to yield overconfident results outside the training data (Fig. 4). Thus, similar to Kristiadi et al. [2], we can optionally add an auxiliary

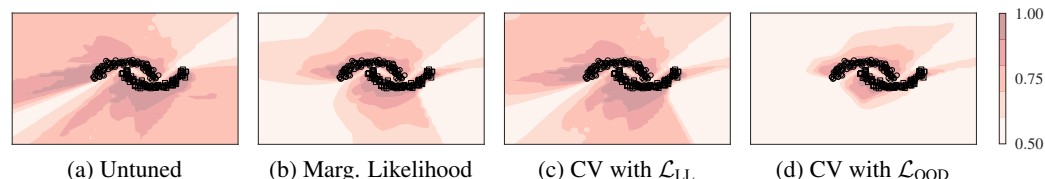

| (a) Untuned | (b) Marg. Likelihood | (c) CV with $\mathcal{L}_{\text{LL}}$ | (d) CV with $\mathcal{L}_{\text{OOD}}$ |

Figure 4: Different objectives for tuning $\boldsymbol{\sigma}^2$. Shades are predictive confidence. Untuned $\boldsymbol{\sigma}^2 = (1, \dots, 1)$ in **(a)**. $\mathcal{D}_{\text{out}}$ consists of uniform noise images.

term to $\mathcal{L}_{\text{LL}}$ that depends on some OOD dataset $\mathcal{D}_{\text{out}}$, resulting in $\mathcal{L}_{\text{OOD}}(\boldsymbol{\sigma}^2) := \mathcal{L}_{\text{LL}}(\boldsymbol{\sigma}^2) + \lambda/C \sum_{\boldsymbol{x}_* \in \mathcal{D}_{\text{out}}} \sum_{c=1}^{C} \log p(y = c \mid \boldsymbol{x}_*, \mathcal{D}; \boldsymbol{\sigma}^2)$. In particular, the additional term is simply the negative cross-entropy between the predictive distribution and the uniform probability vector of length $C$, with $\lambda = 0.5$ as proposed by Hendrycks et al. [18]. Note that both objectives can be optimized via gradient descent without the need of backprop through the network. See Fig. 4 for comparison between different objectives. In Section 5, we discuss the choice of $\mathcal{D}_{\text{out}}$.

## 4  Related work

Mitigation of asymptotic overconfidence has been studied recently: Hein et al. [3] noted, demonstrated, and analyzed this issue, but their proposed method does not work for large $\alpha$. Kristiadi et al. [2] showed that a Bayesian treatment could mitigate this issue even as $\alpha \to \infty$. However, their analysis is restricted to binary classification and the asymptotic confidence of standard ReLU BNNs only converges to a constant in $(0, 1)$. In a non-Bayesian framework, Meinke and Hein [8] used density estimation to achieve the uniform confidence far away from the data. Nevertheless, this property has not been previously achieved in the context of BNNs.

Unlike a line of works that connects NNs and GPs [19–21, etc.] which studies properties of NNs as GPs in an infinite-width limit, we focus on combining *finite-width* BNNs with a GP *a posteriori*. Though similar in spirit, our method thus differs from Wilson et al. [22] which propose a combination of a weight-space prior and a function-space posterior for efficient GP posterior sampling. Our method is also distinct from other methods that model the residual of a predictive model with a GP [5, 15, 6, 7, etc.] since RGPR models the *uncertainty residual* of BNNs, in contrast to the predictive residual of point-estimated networks, and RGPR does not require further posterior inference given a pre-trained BNN.

Cho and Saul [19] proposed a family of kernels for deep learning, called the arc-cosine kernels. The first-order arc-cosine kernel can be interpreted as a ReLU kernel but it only has a quadratic variance growth and thus is not suitable to guarantee the uniform asymptotic confidence. While higher-order arc-cosine kernels have super-quadratic variance growth, they ultimately cannot be interpreted as ReLU kernels, and hence are not as natural as the cubic-spline kernel in the context of ReLU BNNs.

## 5  Empirical Evaluations

We empirically validate Theorem 4 in the asymptotic regime and the effect of RGPR on non-asymptotic confidence estimates in multi-class image classification. The LeNet architecture [23] is used for MNIST, while ResNet-18 [24] is used for CIFAR10, SVHN, and CIFAR100—details in Appendix D. For each dataset, we tune $\boldsymbol{\sigma}^2$ via a validation set of size 2000 obtained by splitting the corresponding test set. Following Hein et al. [3], $\mathcal{D}_{\text{out}}$ consists of smoothed noise images, which are obtained via random permutation, blurring, and contrast rescaling of the original dataset—they do not preserve the structure of the original images and thus can be considered as synthetic noise images. Particularly for ResNet, we use the outputs of its residual blocks to obtain input representations $\boldsymbol{h}_*$.

### 5.1  Asymptotic Regime

In this experiment, we use the last-layer Laplace approximation (LLL) as the base BNN, which has previously been shown to be a strong baseline [2]. Results with other, more sophisticated BNNs

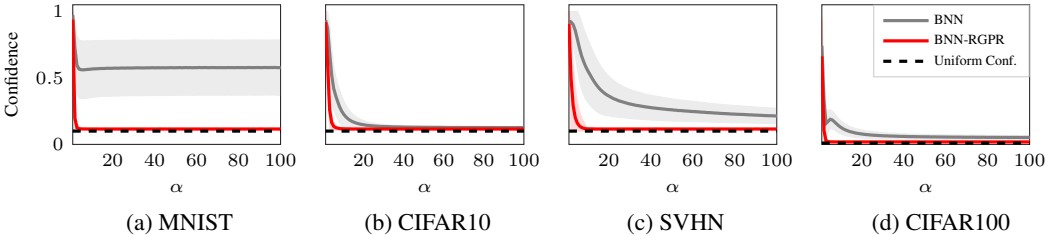

(a) MNIST      (b) CIFAR10      (c) SVHN      (d) CIFAR100

Figure 5: Confidence of a vanilla BNN (LLL) and the same BNN with RGPR, as a function of $\alpha$. Test data are constructed by scaling the original test set. Curves are means, shades are $\pm 1$ std. devs. Note that in **(b)** and **(d)**, even though close, the BNN does not achieve the uniform confidence.

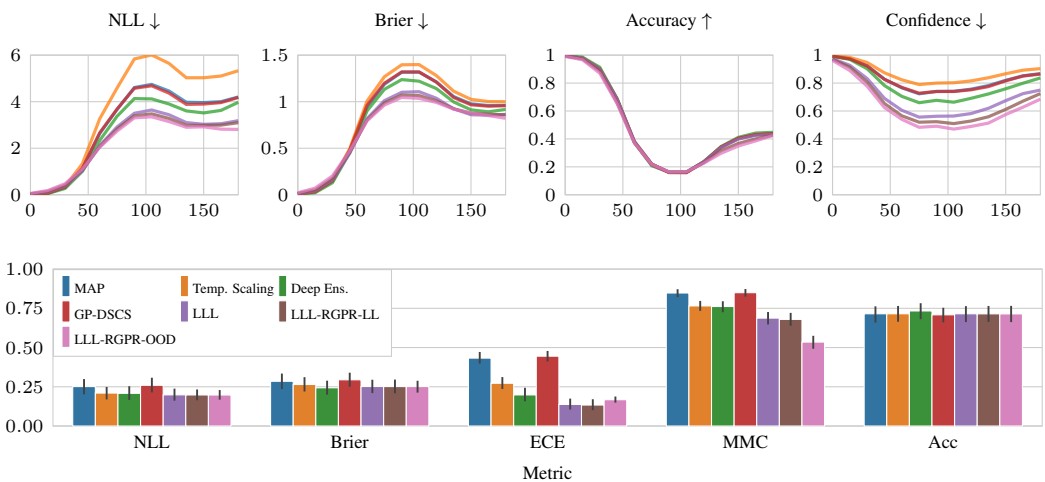

Figure 6: **(Top)** Rotated-MNIST ($x$-axes are rotation angles). **(Bottom)** Corrupted-CIFAR10—values are normalized to $[0, 1]$ and are averages over all types of corruption and all severity levels.

[25–27] are in Appendix D—we observe similar results there. Figure 5 shows confidence estimates of both the BNN and the RGPR-imbued BNN over 1000 samples obtained from each of MNIST, CIFAR10, SVHN, and CIFAR100 test sets, as the scaling factor $\alpha$ increases. As expected, the vanilla BNN does not achieve the ideal uniform confidence prediction, even for large $\alpha$. This issue is most pronounced on MNIST, where the confidence estimates are far away from the ideal confidence of 0.1. Overall, this observation validates the hypothesis that BNNs have residual uncertainty, leading to asymptotic overconfidence that can be severe. We confirm that RGPR fixes this issue. Moreover, its convergence appears at a finite, small $\alpha$; without a pronounced effect on the original confidence.

## 5.2 Non-Asymptotic Regime

We report results on standard dataset shift and out-of-distribution (OOD) detection tasks. For the former, we use the standard rotated-MNIST and CIFAR10-C datasets [28, 29] and measure the performance using the following metrics: negative log-likelihood (NLL), the Brier score, expected calibration error (ECE), accuracy, and average confidence. Meanwhile, for OOD detection, we use five OOD sets for each in-distribution dataset. The FPR@95 metric measures the false positive rate of an OOD detector at a 95% true positive rate. We use LLL

Table 1: OOD data detection in terms of FPR@95. All values are in percent and averages over five OOD test sets and over 5 prediction runs.

| Methods | MNIST | CIFAR10 | SVHN | CIFAR100 |
|---|---|---|---|---|
| MAP | 28.2 | 38.9 | 17.8 | 72.2 |
| TS | 28.4 | 34.9 | 17.6 | 71.9 |
| DE | 23.0 | 51.0 | 11.3 | 74.7 |
| GP-DSCS | 27.8 | 46.7 | 19.1 | 69.1 |
| LLL | 24.8 | 29.8 | 15.7 | 69.5 |
| LLL-RGPR-LL | 3.9 | 29.6 | 13.8 | 65.8 |
| LLL-RGPR-OOD | **3.6** | **24.2** | **9.6** | **63.0** |

as the base BNN for RGPR and compare it against the MAP-trained network, temperature scaling [TS, 30], the method of Qiu et al. [7] with the DSCS kernel (GP-DSCS, see Appendix C), and Deep

Ensemble [DE, 31], which is a strong baseline in this regime [28]. We denote the RGPR tuned via $\mathcal{L}_{\text{LL}}$ and $\mathcal{L}_{\text{OOD}}$ with the suffixes "-LL" and "-OOD", respectively. More results are in Appendix D.

On the rotated-MNIST benchmark, we observe in Fig. 6 that RGPR consistently improves the base LLL, especially when tuned with $\mathcal{L}_{\text{OOD}}$, while still preserving the calibration of LLL on the clean data. LLL-RGPR attains better results than GP-DSCS, which confirms that applying a GP on top of a trained BNN is more effective than on top of MAP-trained nets. Some improvements, albeit less pronounced (see Table 3 in the appendix for the complementary numerical values), are also observed in CIFAR10-C. For OOD detection (Table 1) we find that LLL is already competitive with all baselines, but RGPR can still improve it further, making it better than Deep Ensemble. Further results comparing RGPR to recent non-Bayesian baselines [32, 33] are in Appendix D.

Finally, we discuss the limitation of $\mathcal{L}_{\text{OOD}}$. While the use of additional OOD data in tuning $\boldsymbol{\sigma}^2$ improves both dataset-shift and OOD detection results, it is not without a drawback: $\mathcal{L}_{\text{OOD}}$ induces slightly worse calibration in terms of ECE (Table 6 in Appendix D). This implies that one can somewhat trade the exactness of RGPR (as assumed by Proposition 2) off with better OOD detection. This trade-off is expected to a degree since OOD data are often close to the training data. Hence, the single multiplicative hyperparameter $\sigma_l^2$ of each the DSCS kernel in (10) cannot simultaneously induce high variance on outliers and low variance on the nearby training data. Table 10 (Appendix D) corroborates this: When a $\mathcal{D}_{\text{out}}$ "closer" to the training data (the $32 \times 32$ ImageNet dataset [34]) is used, the ECE values induced by $\mathcal{L}_{\text{OOD}}$ become worse (but the OOD performance improves further). Note that this negative correlation between ECE and OOD detection performance also presents in state-of-the-art OOD detectors (Section D.3.5). So, if the in-distribution calibration performance is more crucial in applications of interest, $\mathcal{L}_{\text{LL}}$ is a better choice for tuning $\boldsymbol{\sigma}^2$ since it still gives benefits on non-asymptotic outliers, but preserves calibration better than $\mathcal{L}_{\text{OOD}}$.

## 6 Conclusion

Extending finite ReLU BNNs with an infinite set of additional, carefully placed ReLU features fixes their asymptotic overconfidence. We do so by generalizing the classic cubic spline kernel, which, when used in a GP prior, yields a marginal variance growing cubically in the distance between a test point and the training data. The simplicity of our method is its main strength: RGPR causes no additional overhead during BNNs' training, but nevertheless meaningfully approximates a full GP posterior, because the proposed kernel contributes only negligible prior variance near the training data. RGPR can thus be applied *post-hoc* to any pre-trained ReLU BNN and causes only a small overhead during prediction. We also showed how RGPR can be extended further—again in a *post-hoc* manner—to also correct the BNN's uncertainty *near* the training data, by modeling residuals in the higher layers of the network. The intuition behind RGPR is relatively simple, but it bridges the domains of deep learning and non-parametric/kernel models: Correctly modeling uncertainty across the input domain requires a non-parametric model of infinitely many ReLU features, but only finitely many such features need to be trained to make good point predictions.

## Acknowledgments and Disclosure of Funding

The authors gratefully acknowledge financial support by the European Research Council through ERC StG Action 757275 / PANAMA; the DFG Cluster of Excellence "Machine Learning - New Perspectives for Science", EXC 2064/1, project number 390727645; the German Federal Ministry of Education and Research (BMBF) through the Tübingen AI Center (FKZ: 01IS18039A); and funds from the Ministry of Science, Research and Arts of the State of Baden-Württemberg. AK is grateful to the International Max Planck Research School for Intelligent Systems (IMPRS-IS) for support. AK is also grateful to Felix Dangel, Jonathan Wenger, Nathanael Bosch, Runa Eschenhagen, Christian Fröhlich, and other members of the Methods of Machine Learning group for feedback.

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
