# An Infinite-Feature Extension for Bayesian ReLU Nets That Fixes Their Asymptotic Overconfidence

## Appendix A    The Cubic Spline Kernel

Recall that we have a linear model $f : [c_{\min}, c_{\max}] \times \mathbb{R}^K \to \mathbb{R}$ with the ReLU feature map $\phi$ defined by $f(x; \boldsymbol{w}) := \boldsymbol{w}^\top \phi(x)$ over the input space $[c_{\min}, c_{\max}] \subset \mathbb{R}$, where $c_{\min} < c_{\max}$. Furthermore, $\phi$ regularly places the $K$ generalized ReLU functions centered at $(c_i)_{i=1}^K$ where $c_i = c_{\min} + \frac{i-1}{K-1}(c_{\max} - c_{\min})$ in the input space, and we consider a Gaussian prior $p(\boldsymbol{w}) := \mathcal{N}\left(\boldsymbol{w} \,\middle|\, \boldsymbol{0}, \sigma^2 K^{-1}(c_{\max} - c_{\min})\boldsymbol{I}\right)$ over the weight $\boldsymbol{w}$. Then, as $K$ goes to infinity, the distribution over the function output $f(x)$ is a Gaussian process with mean 0 and covariance

$$\mathrm{cov}(f(x), f(x')) = \sigma^2 \frac{c_{\max} - c_{\min}}{K} \phi(x)^\top \phi(x') = \sigma^2 \frac{c_{\max} - c_{\min}}{K} \sum_{i=1}^K \mathrm{ReLU}(x; c_i)\mathrm{ReLU}(x'; c_i)$$

$$= \sigma^2 \frac{c_{\max} - c_{\min}}{K} \sum_{i=1}^K H(x - c_i)H(x' - c_i)(x - c_i)(x' - c_i)$$

$$= \sigma^2 \frac{c_{\max} - c_{\min}}{K} \sum_{i=1}^K H(\min(x, x') - c_i)\left(c_i^2 - c_i(x + x') + xx'\right), \tag{11}$$

where the last equality follows from (i) the fact that both $x$ and $x'$ must be greater than or equal to $c_i$, and (ii) by expanding the quadratic form in the second line.

Let $\bar{x} := \min(x, x')$. Since (11) is a Riemann sum, in the limit of $K \to \infty$, it is expressed by the following integral

$$\lim_{K \to \infty} \mathrm{cov}(f(x), f(x')) = \sigma^2 \int_{c_{\min}}^{c_{\max}} H(\bar{x} - c)\left(c^2 - c(x + x') + xx'\right) dc$$

$$= \sigma^2 H(\bar{x} - c_{\min}) \int_{c_{\min}}^{\min\{\bar{x}, c_{\max}\}} c^2 - c(x + x') + xx' \, dc$$

$$= \sigma^2 H(\bar{x} - c_{\min})\left[\frac{1}{3}(z^3 - c_{\min}^3) - \frac{1}{2}(z^2 - c_{\min}^2)(x + x') + (z - c_{\min})xx'\right]$$

where we have defined $z := \min\{\bar{x}, c_{\max}\}$. The term $H(\bar{x} - c_{\min})$ has been added in the second equality as the previous expression is zero if $\bar{x} \leq c_{\min}$ (since in this region, all the ReLU functions evaluate to zero). Note that

$$H(\bar{x} - c_{\min}) = H(x - c_{\min})H(x' - c_{\min})$$

is itself a positive definite kernel. We also note that $c_{\max}$ can be chosen sufficiently large so that $[-c_{\max}, c_{\max}]^d$ contains the data for sure, e.g. this is anyway true for data from bounded domains like images in $[0, 1]^d$, and thus we can set $z = \bar{x} = \min(x, x')$.

## Appendix B    Proofs

**Lemma 1.** *Let $0 < \delta < 1$, and let $\sigma^2 > 0$ be a constant. For any $\boldsymbol{x}, \boldsymbol{x}' \in \mathbb{R}^N$ with $\|\boldsymbol{x}\|^2, \|\boldsymbol{x}'\|^2 \leq \delta$ we have $k(\boldsymbol{x}, \boldsymbol{x}'; \sigma^2) \in O(\delta^3)$.*

*Proof.* First, note that $\|\boldsymbol{x}\|^2, \|\boldsymbol{x}'\|^2 \leq \delta$ implies $x_i, x_i' \leq \delta$ for all $i = 1, \ldots, N$. By definition of the 1D DSCS kernel $\overrightarrow{k}^1(x_i, x_i'; \sigma^2)$, it is upper bounded by $\sigma^2(\frac{1}{3}\delta^3)$ since $\bar{x}_i = \min(x_i, x_i') \leq \delta$; and similarly for $\overleftarrow{k}^1(x_i, x_i'; \sigma^2)$ by the symmetry of the DSCS kernel. Thus $k^1(x_i, x_i'; \sigma^2) \in O(\delta^3)$ and hence $k(\boldsymbol{x}, \boldsymbol{x}'; \sigma^2)$ also is, since it is just the average of $\{k^1(x_i, x_i'; \sigma^2)\}_{i=1}^N$. □

Before we begin to prove Proposition 2, we need the following lemma by Higham [35]. This lemma is useful to show the approximation errors in (6) and (7).

**Lemma 5 (Higham, 1994).** *Let $\boldsymbol{A}\boldsymbol{m} = \boldsymbol{b}$ and $(\boldsymbol{A} + \Delta\boldsymbol{A})\boldsymbol{n} = \boldsymbol{b} + \Delta\boldsymbol{b}$, and let $\boldsymbol{E}$ and $\boldsymbol{d}$ be a matrix and vector with non-negative components, respectively. Assume that $\|\Delta\boldsymbol{A}\| \leq \epsilon\|\boldsymbol{E}\|$ and $\|\Delta\boldsymbol{b}\| \leq \epsilon\|\boldsymbol{d}\|$, and that $\epsilon\|\boldsymbol{A}^{-1}\|\|\boldsymbol{E}\| < 1$, where $\epsilon > 0$. Then*

$$\|\boldsymbol{m} - \boldsymbol{n}\| \leq \frac{\epsilon(\|\boldsymbol{A}^{-1}\|\|\boldsymbol{d}\| + \|\boldsymbol{m}\|\|\boldsymbol{A}^{-1}\|\|\boldsymbol{E}\|)}{1 - \epsilon\|\boldsymbol{A}^{-1}\|\|\boldsymbol{E}\|}. \tag{12}$$

$\square$

**Proposition 2 (RGPR's GP Posterior).** *Let $f : \mathbb{R}^N \times \mathbb{R}^D \to \mathbb{R}$ be a ReLU BNN with weight distribution $\mathcal{N}(\boldsymbol{\theta} \mid \boldsymbol{\mu}, \boldsymbol{\Sigma})$, and let $\mathcal{D} := (\boldsymbol{x}_m, y_m)_{m=1}^M =: (\boldsymbol{X}, \boldsymbol{y})$ be a dataset. Assume that $\|\boldsymbol{x}_m\|^2, \|\boldsymbol{x}\|^2 \leq \delta$ for all $m = 1, \ldots, M$ and any i.i.d. test point $\boldsymbol{x} \in \mathbb{R}^N$, with $0 < \delta < 1$. Then given an i.i.d. input point $\boldsymbol{x}_* \in \mathbb{R}^N$, under the linearization of $f$ w.r.t. $\boldsymbol{\theta}$ around $\boldsymbol{\mu}$, the GP posterior over $\widetilde{f}_*$ is a Gaussian with mean and variance*

$$\mathbb{E}(\widetilde{f}_* \mid \mathcal{D}) \approx f(\boldsymbol{x}_*; \boldsymbol{\mu}) + \boldsymbol{h}_*^\top \boldsymbol{C}^{-1}(\boldsymbol{y} - f(\boldsymbol{X}; \boldsymbol{\mu})), \tag{6}$$

$$\mathrm{Var}(\widetilde{f}_* \mid \mathcal{D}) \approx \boldsymbol{g}(\boldsymbol{x}_*)^\top \boldsymbol{\Sigma}\boldsymbol{g}(\boldsymbol{x}_*) + k(\boldsymbol{x}_*, \boldsymbol{x}_*) - \boldsymbol{h}_*^\top \boldsymbol{C}^{-1}\boldsymbol{h}_*, \tag{7}$$

*respectively, where $\boldsymbol{h}_* := (\mathrm{Cov}(f(\boldsymbol{x}_*), f(\boldsymbol{x}_1)), \ldots, \mathrm{Cov}(f(\boldsymbol{x}_*), f(\boldsymbol{x}_M)))^\top$, while $\boldsymbol{C}$ is the covariance matrix $(\mathrm{Cov}(f(\boldsymbol{x}_i), f(\boldsymbol{x}_j)))_{ij}^M$, and $f(\boldsymbol{X}; \boldsymbol{\mu}) := (f(\boldsymbol{x}_1; \boldsymbol{\mu}), \ldots, f(\boldsymbol{x}_M; \boldsymbol{\mu}))^\top$. Moreover, the approximation error in (6) is in $O\left((\delta^6\|\boldsymbol{C}^{-1}\|\|\boldsymbol{m}\|)/(1 - \delta^3\|\boldsymbol{C}^{-1}\|)\right)$ where $\boldsymbol{m} = \boldsymbol{C}^{-1}(\boldsymbol{y} - f(\boldsymbol{X}; \boldsymbol{\mu}))$, while the error in (7) is in $O\left((\delta^6(\|\boldsymbol{C}^{-1}\| + \|\boldsymbol{C}^{-1}\|\|\boldsymbol{m}\|))/(1 - \delta^3\|\boldsymbol{C}^{-1}\|)\right)$ where $\boldsymbol{m} = \boldsymbol{C}^{-1}\boldsymbol{h}_*$.*

*Proof.* Under the linearization of $f$ w.r.t. $\boldsymbol{\theta}$ around $\boldsymbol{\mu}$, we have

$$f(\boldsymbol{x}; \boldsymbol{\theta}) \approx f(\boldsymbol{x}; \boldsymbol{\mu}) + \underbrace{\nabla_{\boldsymbol{\theta}} f(\boldsymbol{x}; \boldsymbol{\theta})|_{\boldsymbol{\mu}}}_{=:\boldsymbol{g}(\boldsymbol{x})}^\top (\boldsymbol{\theta} - \boldsymbol{\mu}).$$

So, the distribution over the function output $f(\boldsymbol{x})$, where $\boldsymbol{\theta}$ has been marginalized out, is given by $f(\boldsymbol{x}) \sim \mathcal{N}(f(\boldsymbol{x}; \boldsymbol{\mu}), \boldsymbol{g}(x)^\top \boldsymbol{\Sigma}\boldsymbol{g}(x))$—see e.g. Bishop [36, Sec. 5.7.3]. The definition of RGPR in (5) thus implies that

$$\widetilde{f}(\boldsymbol{x}) \sim \mathcal{N}(f(\boldsymbol{x}; \boldsymbol{\mu}), \boldsymbol{g}(\boldsymbol{x})^\top \boldsymbol{\Sigma}\boldsymbol{g}(\boldsymbol{x}) + k(\boldsymbol{x}, \boldsymbol{x})),$$

since $\widetilde{f}(\boldsymbol{x})$ is a sum of two Normal r.v.s. Note that we can see this distribution as a marginal distribution of a Gaussian process with a mean function $f(\cdot; \boldsymbol{\mu})$ and a kernel $(\boldsymbol{x}, \boldsymbol{x}') \mapsto \boldsymbol{g}(\boldsymbol{x})^\top \boldsymbol{\Sigma}\boldsymbol{g}(\boldsymbol{x}') + k(\boldsymbol{x}, \boldsymbol{x}')$. Thus, we write the following GP prior

$$\widetilde{f}(\boldsymbol{x}) \sim \mathcal{GP}(f(\boldsymbol{x}; \boldsymbol{\mu}), \underbrace{\boldsymbol{g}(\boldsymbol{x})^\top \boldsymbol{\Sigma}\boldsymbol{g}(\boldsymbol{x}') + k(\boldsymbol{x}, \boldsymbol{x}')}_{=:\overline{k}(\boldsymbol{x}, \boldsymbol{x}')}).$$

Our goal is to find the corresponding GP posterior under the dataset $\mathcal{D}$.

Let $\boldsymbol{x}_* \in \mathbb{R}^N$ be an arbitrary test point. The GP posterior at $\boldsymbol{x}_*$, i.e. the predictive distribution of $\widetilde{f}_* := f(\boldsymbol{x}_*)$, is thus identified by the following mean and variance (see e.g. [17]):

$$\mathbb{E}(\widetilde{f}_* \mid \mathcal{D}) = f(\boldsymbol{x}_*; \boldsymbol{\mu}) + \overline{k}(\boldsymbol{x}_*, \boldsymbol{X})^\top \overline{k}(\boldsymbol{X}, \boldsymbol{X})^{-1}(\boldsymbol{y} - f(\boldsymbol{X}; \boldsymbol{\mu})) \tag{13}$$

$$\mathrm{Var}(\widetilde{f}_* \mid \mathcal{D}) = \overline{k}(\boldsymbol{x}_*, \boldsymbol{x}_*) - \overline{k}(\boldsymbol{x}_*, \boldsymbol{X})^\top \overline{k}(\boldsymbol{X}, \boldsymbol{X})^{-1}\overline{k}(\boldsymbol{x}_*, \boldsymbol{X}), \tag{14}$$

where we have used the shorthand $\overline{k}(\boldsymbol{x}_*, \boldsymbol{X}) := (\overline{k}(\boldsymbol{x}_*, \boldsymbol{x}_1), \ldots, \overline{k}(\boldsymbol{x}_*, \boldsymbol{x}_M))^\top$ and $\overline{k}(\boldsymbol{X}, \boldsymbol{X})$ is the $M \times M$ kernel matrix of $\overline{k}$ under the training inputs $\boldsymbol{X}$. For the latter we can also write $\overline{k}(\boldsymbol{X}, \boldsymbol{X}) = \boldsymbol{C} + k(\boldsymbol{X}, \boldsymbol{X})$, where $\boldsymbol{C}$ is the kernel matrix of $\boldsymbol{g}(\boldsymbol{x})^\top \boldsymbol{\Sigma}\boldsymbol{g}(\boldsymbol{x}')$ under $\boldsymbol{X}$.

Since we assume $\|\boldsymbol{x}_m\|^2, \|\boldsymbol{x}\|^2 \leq \delta$ for all $m = 1, \ldots, M$ and any i.i.d. test point $\boldsymbol{x} \in \mathbb{R}^N$, we have $k(\boldsymbol{x}, \boldsymbol{x}_m) \approx 0$. Thus, we have $\overline{k}(\boldsymbol{X}, \boldsymbol{X}) \approx \boldsymbol{C}$ and

$$\overline{k}(\boldsymbol{x}_*, \boldsymbol{X}) \approx (\boldsymbol{g}(\boldsymbol{x}_*)^\top \boldsymbol{\Sigma}\boldsymbol{g}(\boldsymbol{x}_1), \ldots, \boldsymbol{g}(\boldsymbol{x}_*)^\top \boldsymbol{\Sigma}\boldsymbol{g}(\boldsymbol{x}_M))^\top$$

$$= (\mathrm{Cov}(f(\boldsymbol{x}_*), f(\boldsymbol{x}_1)), \ldots, \mathrm{Cov}(f(\boldsymbol{x}_*), f(\boldsymbol{x}_1)))^\top = \boldsymbol{h}_*,$$

where the covariances above are of the network's outputs under the linearization. And so the mean and the variance of the GP posterior simplify to

$$\mathbb{E}(\widetilde{f}_* \mid \mathcal{D}) \approx f(\boldsymbol{x}_*; \boldsymbol{\mu}) + \boldsymbol{h}_*^\top \boldsymbol{C}^{-1}(\boldsymbol{y} - f(\boldsymbol{X}; \boldsymbol{\mu}))$$

and

$$\mathrm{Var}(\widetilde{f}_* \mid \mathcal{D}) \approx \boldsymbol{g}(\boldsymbol{x}_*)^\top \boldsymbol{\Sigma} \boldsymbol{g}(\boldsymbol{x}_*) + k(\boldsymbol{x}_*, \boldsymbol{x}_*) - \boldsymbol{h}_*^\top \boldsymbol{C}^{-1} \boldsymbol{h}_*.$$

The only thing that remains is to obtain the approximation errors of both the mean and variance above. Using Lemma 5, we find the error of $(\boldsymbol{C} + k(\boldsymbol{X}, \boldsymbol{X}))^{-1}(\boldsymbol{y} - f(\boldsymbol{X}; \boldsymbol{\mu}))$ in (13) due to RGPR, i.e. we quantify the error caused by $\delta$ presents in $k(\boldsymbol{X}, \boldsymbol{X})$. We set $\boldsymbol{A} = \boldsymbol{C}$, $\Delta \boldsymbol{A} = k(\boldsymbol{X}, \boldsymbol{X})$, and $\boldsymbol{b} = \boldsymbol{y} - f(\boldsymbol{X}; \boldsymbol{\mu})$. Moreover, we set $\boldsymbol{m} = \boldsymbol{C}^{-1}(\boldsymbol{y} - f(\boldsymbol{X}; \boldsymbol{\mu}))$ and $\boldsymbol{n} = (\boldsymbol{C} + k(\boldsymbol{X}, \boldsymbol{X}))^{-1}(\boldsymbol{y} - f(\boldsymbol{X}; \boldsymbol{\mu}))$. For simplicity, we let $\boldsymbol{E} := \boldsymbol{1}\boldsymbol{1}^\top$ and set $\epsilon = \delta^3 c$ for some constant $c$ s.t. the conditions in Lemma 5 are satisfied. Note that the $\delta^3$ term in $\epsilon$ is so that the condition $\|\Delta \boldsymbol{A}\| \leq \epsilon \|\boldsymbol{E}\|$ is satisfied, since one can write $\|\Delta \boldsymbol{A}\| = c_0 \|\boldsymbol{E}\|$ where $c_0 \in O(\delta^3)$. Moreover, we set $\boldsymbol{d} = \boldsymbol{0}$ since $\Delta \boldsymbol{b} = \boldsymbol{0}$. Plugging these into (12), we thus have

$$\|\boldsymbol{m} - \boldsymbol{n}\| \in O\left(\frac{\delta^3 \|\boldsymbol{A}^{-1}\| \|\boldsymbol{m}\|}{1 - \delta^3 \|\boldsymbol{A}^{-1}\|}\right).$$

Combining this with the $O(\delta^3)$ error in the approximation $\overline{k}(\boldsymbol{x}_*, \boldsymbol{X}) \approx \boldsymbol{h}_*$, we conclude that using (6) as an approximation of (13) incurs an error of

$$O\left(\frac{\delta^6 \|\boldsymbol{A}^{-1}\| \|\boldsymbol{m}\|}{1 - \delta^3 \|\boldsymbol{A}^{-1}\|}\right),$$

which is small since $\delta \in (0, 1)$.

For the approximation error of the variance, we use $\boldsymbol{A}$, $\Delta \boldsymbol{A}$, $\boldsymbol{E}$, and $\epsilon$ as before. But, here we set $\boldsymbol{b} = \boldsymbol{h}_*$, $\Delta \boldsymbol{b} = k(\boldsymbol{x}_*, \boldsymbol{X})$, and $\boldsymbol{d} = \boldsymbol{1}$. Moreover, we set $\boldsymbol{m} = \boldsymbol{C}^{-1} \boldsymbol{h}_*$ and $\boldsymbol{n} = (\boldsymbol{C} + k(\boldsymbol{X}, \boldsymbol{X}))^{-1}(\boldsymbol{h}_* + k(\boldsymbol{x}_*, \boldsymbol{X}))$. Then, plugging them into Lemma 5, we obtain

$$\|\boldsymbol{m} - \boldsymbol{n}\| \in O\left(\frac{\delta^3 (\|\boldsymbol{A}^{-1}\| + \|\boldsymbol{A}^{-1}\| \|\boldsymbol{m}\|)}{1 - \delta^3 \|\boldsymbol{A}^{-1}\|}\right).$$

Combining this with the approximation error in $\overline{k}(\boldsymbol{x}_*, \boldsymbol{X}) \approx \boldsymbol{h}_*$ as before, we obtain the desired result. $\qquad\square$

To prove Lemma 3 and Theorem 4, we need the following definition. Let $f : \mathbb{R}^N \times \mathbb{R}^D \to \mathbb{R}^C$ defined by $(\boldsymbol{x}, \boldsymbol{\theta}) \mapsto f(\boldsymbol{x}; \boldsymbol{\theta})$ be a feed-forward neural network which uses piecewise-affine activation functions (such as ReLU and leaky-ReLU) and are linear in the output layer. Such a network is called a ***ReLU network*** and can be written as a continuous piecewise-affine function [37]. That is, there exists a finite set of polytopes $\{Q_i\}_{i=1}^P$—referred to as ***linear regions*** $f$—such that $\cup_{i=1}^P Q_i = \mathbb{R}^N$ and $f|_{Q_i}$ is an affine function for each $i = 1, \ldots, P$ [3]. The following lemma is central in our proofs below (the proof is in Lemma 3.1 of Hein et al. [3]).

**Lemma 6** (Hein et al., 2019). *Let $\{Q_i\}_{i=1}^P$ be the set of linear regions associated to the ReLU network $f : \mathbb{R}^N \times \mathbb{R}^D \to \mathbb{R}^C$, For any $\boldsymbol{x} \in \mathbb{R}^N$ with $\boldsymbol{x} \neq 0$ there exists a positive real number $\beta$ and $j \in \{1, \ldots, P\}$ such that $\alpha \boldsymbol{x} \in Q_j$ for all $\alpha \geq \beta$.* $\qquad\square$

**Lemma 3** (**Asymptotic Variance Growth**). *Let $f : \mathbb{R}^N \times \mathbb{R}^D \to \mathbb{R}^C$ be a pre-trained ReLU network with posterior $\mathcal{N}(\boldsymbol{\theta} \mid \boldsymbol{\mu}, \boldsymbol{\Sigma})$ and $\widetilde{f}$ be obtained from $f$ via RGPR. Suppose that the linearization of $f$ w.r.t. $\boldsymbol{\theta}$ around $\boldsymbol{\mu}$ is employed. For any $\boldsymbol{x}_* \in \mathbb{R}^N$ with $\boldsymbol{x}_* \neq 0$ there exists $\beta > 0$ such that for any $\alpha \geq \beta$ and each $c = 1, \ldots, C$, the variance $\mathrm{Var}(\widetilde{f}^{(c)}(\alpha \boldsymbol{x}_*))$ under (9) is in $\Theta(\alpha^3)$.*

*Proof.* Let $\boldsymbol{x}_* \in \mathbb{R}^N$ with $\boldsymbol{x}_* \neq 0$ be arbitrary. By Lemma 6 and definition of ReLU network, there exists a linear region $R$ and real number $\beta > 0$ such that for any $\alpha \geq \beta$, the restriction of $f$ to $R$ can be written as

$$f|_R(\alpha \boldsymbol{x}; \boldsymbol{\theta}) = \boldsymbol{W}(\alpha \boldsymbol{x}) + \boldsymbol{b},$$

for some matrix $\boldsymbol{W} \in \mathbb{R}^{C \times N}$ and vector $\boldsymbol{b} \in \mathbb{R}^C$, which are functions of the parameter $\boldsymbol{\theta}$, evaluated at $\boldsymbol{\mu}$. In particular, for each $c = 1, \ldots, C$, the $c$-th output component of $f|_R$ can be written as

$$f_c|_R = \boldsymbol{w}_c^\top (\alpha \boldsymbol{x}) + b_c,$$

where $\boldsymbol{w}_c$ and $b_c$ are the $c$-th row of $\boldsymbol{W}$ and $\boldsymbol{b}$, respectively.

Let $c \in \{1, \ldots, C\}$ and let $\boldsymbol{j}_c(\alpha \boldsymbol{x}_*)$ be the $c$-th column of the Jacobian $\boldsymbol{J}(\alpha \boldsymbol{x}_*)$ as defined in (1). Then by definition of $p(\widetilde{f}_* \mid \boldsymbol{x}_*, \mathcal{D})$, the variance of $\widetilde{f}_c|_R(\alpha \boldsymbol{x}_*)$—the $c$-th diagonal entry of the covariance of $p(\widetilde{f}_* \mid \boldsymbol{x}_*, \mathcal{D})$—is given by

$$\mathrm{var}(\widetilde{f}_c|_R(\alpha \boldsymbol{x}_*)) = \boldsymbol{j}_c(\alpha \boldsymbol{x}_*)^\top \boldsymbol{\Sigma} \boldsymbol{j}_c(\alpha \boldsymbol{x}_*) + k(\alpha \boldsymbol{x}_*, \alpha \boldsymbol{x}_*).$$

Now, from the definition of the DSCS kernel in (4), we have

$$k(\alpha \boldsymbol{x}_*, \alpha \boldsymbol{x}_*) = \frac{1}{N} \sum_{i=1}^{N} k^1(\alpha x_{*i}, \alpha x_{*i}) = \frac{1}{N} \sum_{i=1}^{N} \alpha^3 \frac{\sigma^2}{3} x_{*i}^3 = \frac{\alpha^3}{N} \sum_{i=1}^{N} k^1(x_{*i}, x_{*i}) \in \Theta(\alpha^3).$$

Furthermore, we have

$$\boldsymbol{j}_c(\alpha \boldsymbol{x}_*)^\top \boldsymbol{\Sigma} \boldsymbol{j}_c(\alpha \boldsymbol{x}_*) = \left( \alpha (\nabla_{\boldsymbol{\theta}} \boldsymbol{w}_c|_{\boldsymbol{\mu}})^\top \boldsymbol{x} + \nabla_{\boldsymbol{\theta}} b_c|_{\boldsymbol{\mu}} \right)^\top \boldsymbol{\Sigma} \left( \alpha (\nabla_{\boldsymbol{\theta}} \boldsymbol{w}_c|_{\boldsymbol{\mu}})^\top \boldsymbol{x} + \nabla_{\boldsymbol{\theta}} b_c|_{\boldsymbol{\mu}} \right).$$

Thus, $\boldsymbol{j}_c(\alpha \boldsymbol{x}_*)^\top \boldsymbol{\Sigma} \boldsymbol{j}_c(\alpha \boldsymbol{x}_*)$ is a quadratic function of $\alpha$. Therefore, $\mathrm{var}(\widetilde{f}_c|_R(\alpha \boldsymbol{x}_*))$ is in $\Theta(\alpha^3)$. $\qquad \square$

**Theorem 4 (Uniform Asymptotic Confidence).** *Let $f : \mathbb{R}^N \times \mathbb{R}^D \to \mathbb{R}^C$ be a $C$-class pretrained ReLU network equipped with the posterior $\mathcal{N}(\boldsymbol{\theta} \mid \boldsymbol{\mu}, \boldsymbol{\Sigma})$ and let $\widetilde{f}$ be obtained from $f$ via RGPR. Suppose that the linearization of $f$ and the generalized probit approximation (2) is used for approximating the predictive distribution $p(y_* = c \mid \alpha \boldsymbol{x}_*, \widetilde{f}, \mathcal{D})$ under $\widetilde{f}$. For any input $\boldsymbol{x}_* \in \mathbb{R}^N$ with $\boldsymbol{x}_* \neq \boldsymbol{0}$ and for every class $c = 1, \ldots, C$, we have $\lim_{\alpha \to \infty} p(y_* = c \mid \alpha \boldsymbol{x}_*, \widetilde{f}, \mathcal{D}) = 1/C$.*

*Proof.* Let $\boldsymbol{x}_* \neq \boldsymbol{0} \in \mathbb{R}^N$ be arbitrary. By Lemma 6 and definition of ReLU network, there exists a linear region $R$ and real number $\beta > 0$ such that for any $\alpha \geq \beta$, the restriction of $f$ to $R$ can be written as

$$f|_R(\alpha \boldsymbol{x}) = \boldsymbol{W}(\alpha \boldsymbol{x}) + \boldsymbol{b},$$

where the matrix $\boldsymbol{W} \in \mathbb{R}^{C \times N}$ and vector $\boldsymbol{b} \in \mathbb{R}^C$ are functions of the parameter $\boldsymbol{\theta}$, evaluated at $\boldsymbol{\mu}$. Furthermore, for $i = 1, \ldots, C$ we denote the $i$-th row and the $i$-th component of $\boldsymbol{W}$ and $\boldsymbol{b}$ as $\boldsymbol{w}_i$ and $b_i$, respectively. Under the linearization of $f$, the marginal distribution (9) over the output $\widetilde{f}(\alpha \boldsymbol{x})$ holds. Hence, under the generalized probit approximation, the predictive distribution restricted to $R$ is given by

$$
\begin{aligned}
\widetilde{p}(y_* = c \mid \alpha \boldsymbol{x}_*, \mathcal{D}) &\approx \frac{\exp(m_c(\alpha \boldsymbol{x}_*) \, \kappa_c(\alpha \boldsymbol{x}_*))}{\sum_{i=1}^{C} \exp(m_i(\alpha \boldsymbol{x}_*) \, \kappa_i(\alpha \boldsymbol{x}_*))} \\
&= \frac{1}{1 + \sum_{i \neq c}^{C} \exp(\underbrace{m_i(\alpha \boldsymbol{x}_*) \, \kappa_i(\alpha \boldsymbol{x}_*) - m_c(\alpha \boldsymbol{x}_*) \, \kappa_c(\alpha \boldsymbol{x}_*)}_{=: z_{ic}(\alpha \boldsymbol{x}_*)})},
\end{aligned}
$$

where for all $i = 1, \ldots, C$,

$$m_i(\alpha \boldsymbol{x}_*) = f_i|_R(\alpha \boldsymbol{x}; \boldsymbol{\mu}) = \boldsymbol{w}_i^\top (\alpha \boldsymbol{x}) + b_i \in \mathbb{R},$$

and

$$\kappa_i(\alpha \boldsymbol{x}) = \left( 1 + \pi/8 \left( v_{ii}(\alpha \boldsymbol{x}_*) + k(\alpha \boldsymbol{x}_*, \alpha \boldsymbol{x}_*) \right) \right)^{-\frac{1}{2}} \in \mathbb{R}_{>0}.$$

In particular, for all $i = 1, \ldots, C$, note that $m(\alpha \boldsymbol{x}_*)_i \in \Theta(\alpha)$ and $\kappa(\alpha \boldsymbol{x})_i \in \Theta(1/\alpha^{\frac{3}{2}})$ since $v_{ii}(\alpha \boldsymbol{x}_*) + k(\alpha \boldsymbol{x}_*, \alpha \boldsymbol{x}_*)$ is in $\Theta(\alpha^3)$ by Lemma 3. Now, notice that for any $c = 1, \ldots, C$ and any $i \in \{1, \ldots, C\} \setminus \{c\}$, we have

$$
\begin{aligned}
z_{ic}(\alpha \boldsymbol{x}_*) &= (m_i(\alpha \boldsymbol{x}_*) \, \kappa_i(\alpha \boldsymbol{x}_*)) - (m_c(\alpha \boldsymbol{x}_*) \, \kappa_c(\alpha \boldsymbol{x}_*)) \\
&= (\underbrace{\kappa_i(\alpha \boldsymbol{x}_*) \, \boldsymbol{w}_i}_{\Theta\left(1/\alpha^{\frac{3}{2}}\right)} - \underbrace{\kappa_c(\alpha \boldsymbol{x}_*) \, \boldsymbol{w}_c}_{\Theta\left(1/\alpha^{\frac{3}{2}}\right)})^\top (\alpha \boldsymbol{x}_*) + \underbrace{\kappa_i(\alpha \boldsymbol{x}_*) \, b_i}_{\Theta\left(1/\alpha^{\frac{3}{2}}\right)} - \underbrace{\kappa_c(\alpha \boldsymbol{x}_*) \, b_c}_{\Theta\left(1/\alpha^{\frac{3}{2}}\right)}.
\end{aligned}
$$

Thus, it is easy to see that $\lim_{\alpha \to \infty} z_{ic}(\alpha \boldsymbol{x}_*) = 0$. Hence we have

$$\lim_{\alpha \to \infty} \widetilde{p}(y_* = c \mid \alpha \boldsymbol{x}_*, \mathcal{D}) = \lim_{\alpha \to \infty} \frac{1}{1 + \sum_{i \neq c}^{C} \exp(z_{ic}(\alpha \boldsymbol{x}_*))} = \frac{1}{1 + \sum_{i \neq c}^{C} \exp(0)} = \frac{1}{C},$$

as required. $\qquad\qquad\qquad\qquad\qquad\qquad\qquad\qquad\qquad\qquad\qquad\qquad\qquad\qquad\qquad\qquad\qquad\square$

## Appendix C    Modeling Residuals with GPs

The method of Blight and Ott [5], henceforth called BNO, models the residual of polynomial regressions. That is, suppose $\phi : \mathbb{R} \to \mathbb{R}^D$ is a polynomial basis function defined by $\phi(x) := (1, x, x^2, \ldots, x^{D-1})$, $k$ is an arbitrary kernel, and $\boldsymbol{w} \in \mathbb{R}^D$ is a weight vector, BNO assumes

$$\widetilde{f}(x) := \boldsymbol{w}^\top \phi(x) + \widehat{f}(x), \qquad \text{where } \widehat{f} \sim \mathcal{GP}(0, k).$$

Recently, this method has been extended to neural networks. Qiu et al. [7] apply the same idea—modeling residuals with GPs—to pre-trained networks, resulting in a method called RIO. Suppose that $f_{\boldsymbol{\mu}} : \mathbb{R}^N \to \mathbb{R}$ is a neural-network with a pre-trained, *point-estimated* parameters $\boldsymbol{\mu}$. Their method is defined by

$$\widetilde{f}(\boldsymbol{x}) := f_{\boldsymbol{\mu}}(\boldsymbol{x}) + \widehat{f}(\boldsymbol{x}), \qquad \text{where } \widehat{f} \sim \mathcal{GP}(0, k_{\mathrm{IO}}).$$

The kernel $k_{\mathrm{IO}}$ is a sum of RBF kernels applied on the dataset $\mathcal{D}$ (inputs) and the network's predictions over $\mathcal{D}$ (outputs), hence the name IO—input-output. As in the original Blight and Ott's method, RIO also focuses on modeling predictive residuals and requires GP posterior inference. Suppose that $m(\boldsymbol{x})$ and $v(\boldsymbol{x})$ is the a posteriori marginal mean and variance of the GP, respectively. Then, via standard computations, one can see that even though $f$ is a point-estimated network, $\widetilde{f}$ is a random function, distributed *a posteriori* by

$$\widetilde{f}(\boldsymbol{x}) \sim \mathcal{N}\left(\widetilde{f}_{\boldsymbol{\mu}}(\boldsymbol{x}) + m(\boldsymbol{x}), v(\boldsymbol{x})\right).$$

Thus, BNO and RIO effectively add uncertainty to point-estimated networks. But, there is no guarantee that they preserve the original predictive performance of $f$ since $m$ is in general non-vanishing.

The posterior inference of BNO and RIO can be computationally intensive, depending on the number of training examples $M$: The cost of exact posterior inference is in $\Theta(M^3)$. While it can be alleviated by approximate inference, such as via inducing point methods and stochastic optimizations, the posterior inference requirement can still be a hindrance for the practical adoption of BNO and RIO, especially on large problems.

## Appendix D    Additional Experiments

### D.1    Asymptotic Regime

As a gold standard GP baseline, we compare against the method of Qiu et al. [7] (with our DSCS kernel). We refer to this baseline simply as GP-DSCS. The base methods, which RGPR is implemented on, are the following recently-proposed BNNs: (i) Kronecker-factored Laplace [KFL, 25], (ii) stochastic weight averaging-Gaussian [SWAG, 26], and (iii) stochastic variational deep kernel learning [SVDKL, 27]. All the kernel hyperparameters for RGPR are set to a constant value of $1 \times 10^{-10}$ since we focus on the asymptotic regime. In all cases, MC-integral with 10 posterior samples is used for making predictions. We construct a test dataset artificially by sampling 2000 uniform noises in $[0, 1]^N$ and scale them with a scalar $\alpha = 2000$. The goal is to achieve low confidence over these far-away points.

The results are presented in Table 2. We observe that the RGPR-augmented methods are significantly better than their respective base methods. In particular, their confidence estimates are significantly lower than those of the vanilla methods, becoming closer to the confidence of the gold-standard GP-DSCS baseline. This indicates that RGPR makes BNNs better calibrated in the asymptotic regime.

Table 2: RGPRs compared to their respective base methods on the detection of far-away outliers. Values are average confidences. Error bars are standard errors over three prediction runs. For each dataset, the best value over each vanilla and RGPR-imbued method (e.g. KFL against KFL-RGPR) are in bold.

| Methods | CIFAR10 | SVHN |
|---|---|---|
| GP-DSCS | 22.0±0.2 | 22.1±0.3 |
| KFL | 64.5±0.7 | 63.4±1.5 |
| KFL-RGPR | **29.9**±0.3 | **27.5**±0.0 |
| SWAG | 63.5±1.8 | 50.2±4.2 |
| SWAG-RGPR | **29.3**±0.2 | **27.5**±0.0 |
| SVDKL | 46.4±0.3 | 49.1±0.2 |
| SVDKL-RGPR | **22.0**±0.1 | **22.1**±0.1 |

Table 3: CIFAR10-C results. Values are mean over all corruptions.

| | NLL | ECE | Brier | Confidence | Accuracy |
|---|---|---|---|---|---|
| MAP | 1.066 | 0.226 | 0.402 | 0.887 | 0.739 |
| Temp. | 0.914 | 0.147 | 0.378 | 0.842 | 0.739 |
| DE | 0.909 | 0.110 | **0.354** | 0.840 | **0.752** |
| GP-DSCS | 1.096 | 0.232 | 0.413 | 0.888 | 0.734 |
| LLL | 0.872 | 0.080 | 0.363 | 0.800 | 0.739 |
| LLL-RGPR-LL | 0.870 | **0.079** | 0.363 | 0.796 | 0.738 |
| LLL-RGPR-OOD | **0.869** | 0.095 | 0.363 | **0.717** | 0.738 |

## D.2 Training Details

For LeNet, we use Adam optimizer with an initial learning rate $1 \times 10^{-3}$ while for ResNet, we use SGD with an initial learning rate of $0.1$ and momentum $0.9$. In both cases, the optimization is carried out for 100 epochs using weight decay $5 \times 10^{-4}$ on a single GPU. We also reduce the learning rate by a factor of 10 at epochs 50, 75, and 90. Test accuracies are in Table 6.

## D.3 Non-Asymptotic Regime

### D.3.1 Dataset shift

In Table 3 we present the non-normalized numerical results to complement Fig. 6. RGPR in general improves the vanilla LLL.

### D.3.2 OOD detection

We expand Table 1 in Table 7. In the same table, we additionally show the mean confidence values [38, MMC,]. For CIFAR10, SVHN, and CIFAR100, we test each model against FMNIST (called FMNIST3D) to measure the performance on grayscale OOD images. Finally, we also show the OOD detection performance via additional AUROC and area under precision-recall curve (AUPRC) metrics in Table 8.

Additionally, we compare RGPR with recent non-Bayesian baselines: (i) the Mahalanobis detector [32] and (ii) deterministic uncertainty quantification (DUQ) [33]. Values are taken directly from the original papers—they used the same architecture as in this paper. Table 4 shows that a RGPR-equipped BNN is better than the Mahalanobis detector. Moreover, LLL-RGPR-OOD is competitive to DUQ, but without the drawback of reducing test accuracy.

Table 4: RGPR against recent non-Bayesian baselines. The OOD detection metric is AUROC.

| | CIFAR10 vs. LSUN | CIFAR10 vs. SVHN |
|---|---|---|
| Mahalanobis | 89.2 | 91.5 |
| LLL-RGPR-OOD | **92.6** | **95.8** |

| | Test Acc. | CIFAR10 vs. SVHN |
|---|---|---|
| DUQ ($\lambda = 0$) | 94.2 | 86.1 |
| DUQ ($\lambda = 0.5$) | 93.2 | **92.7** |
| LLL-RGPR-OOD | **94.3** | 92.6 |

Table 5: Expected calibration errors (ECE).

| | MNIST | CIFAR10 | SVHN | CIFAR100 |
|---|---|---|---|---|
| MAP | 6.7 | 13.1 | 10.1 | 8.1 |
| Temp. Scaling | 11.4 | 3.6 | 2.1 | 6.4 |
| ACET | 5.9 | 15.8 | 11.9 | 10.1 |
| OE | 14.7 | 15.8 | 11.0 | 25.0 |

### D.3.3  Hyperparameter tuning

We present the optimal hyperparameters $(\sigma_l^2)_{l=0}^{L-1}$ in Table 9. We observe that using higher representations of the data is beneficial, as indicated by non-trivial hyperparameter values on all layers across all networks and datasets.

### D.3.4  Natural images for tuning

We present OOD detection results via different $\mathcal{D}_{\text{our}}$ for tuning $\boldsymbol{\sigma}^2$, in Table 10. Specifically, we use the ImageNet32×32 dataset [34], which represents natural image datasets, and is thus more sophisticated than the noise dataset used in the main text. Nevertheless, we observe that the OOD detection performance is comparable to that of the noise dataset, justifying the choice of $\mathcal{D}_{\text{out}}$ we have made in the main text.

### D.3.5  Calibration is at odds with OOD detection

As noted in the main text, we observe that employing OOD data for tuning $\boldsymbol{\sigma}^2$ degrades the in-distribution calibration (as measured by the ECE metric) of RGPR. In Table 5 (taken from Table 5 of Kristiadi et al. [2]), we can see that even recent OOD training methods with many more parameters than RGPR such as ACET [3] and OE [18] degrade the in-distribution ECE. However, note that ACET and OE represent state-of-the-art OOD detectors. Hence, it is reasonable to conclude that this issue does not seem to be inherent to RGPR.

### D.4  Regression

To empirically validate our method and analysis (esp. Lemma 3), we present a toy regression results in Fig. 7. RGPR improves the BNN further: Far away from the data, the error bar becomes wider. For more challenging problems, we employ a subset of the standard UCI regression datasets. Our goal here, similar to the classification case, is to compare the uncertainty behavior of RGPR-augmented BNN baselines near the training data (inliers) and far away from them (outliers). The outlier dataset is constructed by sampling 1000 points from the standard Gaussian and scale them with $\alpha = 2000$. The metric used is the predictive error bar (standard deviation), i.e. the same metric visually used in Fig. 7. Following the standard practice (see e.g. Sun et al. [39]), we use a two-layer ReLU network with 50 hidden units. The Bayesian methods used are LLL, KFL, SWAG, and stochastic variational GP [SVGP, 16] using 50 inducing points. Finally, we standardize the data and the hyperparameter for RGPR is set to 0.001 so that Proposition 2 is satisfied. The results are presented in Table 11. We can observe that RGPR retain high confidence estimates over inlier data and yield much larger error bars compared to the base methods.

Table 6: OOD data detection in terms of FPR@95. All values are in percent and averages over five OOD test sets and over 5 prediction runs.

| Methods | MNIST | CIFAR10 | SVHN | CIFAR100 |
|---|---|---|---|---|
| **Acc. ↑** | | | | |
| MAP | 99.4 | 94.3 | 97.1 | 76.7 |
| Temp. Scaling | 99.4 | 94.3 | 97.1 | 76.7 |
| Deep Ens. | 99.6 | 95.3 | 97.4 | 79.5 |
| GP-DSCS | 99.3 | 93.9 | 97.0 | 76.6 |
| LLL | 99.4 | 94.3 | 97.0 | 76.7 |
| LLL-RGPR-LL | 99.2 | 94.4 | 97.0 | 76.7 |
| LLL-RGPR-OOD | 99.1 | 94.3 | 96.9 | 76.6 |
| **ECE ↓** | | | | |
| MAP | 5.4 | 13.9 | 13.3 | 6.4 |
| Temp. Scaling | 9.9 | 6.7 | 7.5 | 4.7 |
| Deep Ens. | 12.5 | 2.8 | 1.3 | 1.9 |
| GP-DSCS | 4.5 | 14.4 | 13.6 | 8.2 |
| LLL | 14.0 | 2.8 | 12.9 | 4.7 |
| LLL-RGPR-LL | 15.8 | 3.6 | 13.1 | 5.7 |
| LLL-RGPR-OOD | 19.6 | 12.5 | 15.9 | 15.8 |

Table 7: OOD data detection results in terms of MMC and FPR@95 metrics. All values are averages and standard errors over 10 prediction trials.

| Datasets | MAP MMC ↓ | MAP FPR ↓ | Temp. Scaling MMC ↓ | Temp. Scaling FPR ↓ | Deep Ens. MMC ↓ | Deep Ens. FPR ↓ | GP-DSCS MMC ↓ | GP-DSCS FPR ↓ | LLL MMC ↓ | LLL FPR ↓ | LLL-RGPR-LL MMC ↓ | LLL-RGPR-LL FPR ↓ | LLL-RGPR-OOD MMC ↓ | LLL-RGPR-OOD FPR ↓ |
|---|---|---|---|---|---|---|---|---|---|---|---|---|---|---|
| **MNIST** | 99.2 | - | 99.5±0.0 | - | 99.1 | - | 99.2±0.0 | - | 97.4±0.0 | - | 97.0±0.0 | - | 96.1±0.0 | - |
| EMNIST | 78.1 | 24.5 | 83.4±0.0 | 24.9±0.0 | 74.1 | **21.4** | 77.6±0.0 | 24.7±0.0 | 62.7±0.0 | 23.3±0.1 | 55.7±0.0 | 21.9±0.1 | **49.4**±0.0 | 21.7±0.1 |
| KMNIST | 73.1 | 14.3 | 79.3±0.0 | 14.1±0.0 | 63.1 | 5.6 | 72.2±0.0 | 13.2±0.0 | 52.7±0.0 | 6.3±0.0 | 17.1±0.0 | 0.4±0.0 | 15.6±0.0 | **0.0**±0.0 |
| FMNIST | 79.8 | 26.8 | 85.0±0.0 | 27.3±0.0 | 71.7 | 11.3 | 79.1±0.0 | 25.5±0.1 | 64.6±0.0 | 19.1±0.2 | 18.1±0.0 | 1.3±0.0 | 15.5±0.0 | **0.0**±0.0 |
| GrayCIFAR10 | 85.7 | 3.6 | 93.4±0.0 | 4.3±0.0 | 72.7 | **0.0** | 85.2±0.0 | 3.5±0.0 | 61.1±0.0 | 0.5±0.0 | **15.1**±0.0 | **0.0**±0.0 | 15.1±0.0 | **0.0**±0.0 |
| UniformNoise | 100.0 | 100.0 | 100.0±0.0 | 100.0±0.0 | 99.9 | 100.0 | 100.0±0.0 | 100.0±0.0 | 95.7±0.0 | 99.7±0.0 | **15.1**±0.0 | **0.0**±0.0 | 15.1±0.0 | **0.0**±0.0 |
| **CIFAR10** | 97.0 | - | 95.0±0.0 | - | 95.6 | - | 96.9±0.0 | - | 93.4±0.0 | - | 93.1±0.0 | - | 85.9±0.0 | - |
| SVHN | 62.5 | 29.3 | 53.7±0.0 | 25.6±0.0 | 59.7 | 37.0 | 69.0±0.0 | 40.0±0.1 | 47.0±0.0 | 24.8±0.1 | 46.7±0.0 | 25.1±0.1 | **40.6**±0.0 | 23.3±0.2 |
| LSUN | 74.5 | 52.7 | 65.9±0.0 | 48.7±0.0 | 65.6 | 50.3 | 76.6±0.0 | 55.1±0.3 | 58.5±0.1 | 44.1±0.7 | 57.4±0.1 | 42.9±0.6 | **48.5**±0.1 | **40.0**±0.5 |
| CIFAR100 | 79.4 | 61.5 | 72.4±0.0 | 59.4±0.0 | 70.7 | 58.0 | 80.0±0.0 | 62.5±0.1 | 66.0±0.0 | 58.2±0.2 | 65.3±0.0 | 58.2±0.2 | **55.6**±0.0 | 54.7±0.2 |
| FMNIST3D | 71.4 | 45.3 | 62.8±0.0 | 41.0±0.0 | 63.0 | 44.1 | 72.6±0.0 | 47.9±0.2 | 53.4±0.0 | 34.7±0.2 | 52.6±0.0 | 34.5±0.2 | **36.6**±0.0 | 16.4±0.3 |
| UniformNoise | 64.7 | 26.2 | 54.7±0.1 | 19.5±0.3 | 73.9 | 86.0 | 75.8±0.1 | 55.3±0.4 | 39.1±0.1 | 2.8±0.1 | 37.9±0.1 | 2.2±0.2 | **32.0**±0.1 | **1.7**±0.3 |
| **SVHN** | 98.5 | - | 97.6±0.0 | - | 97.8 | - | 98.5±0.0 | - | 92.4±0.0 | - | 92.2±0.0 | - | 88.0±0.0 | - |
| CIFAR10 | 70.4 | 18.3 | 64.7±0.0 | 18.0±0.0 | 57.2 | **11.9** | 70.9±0.0 | 19.8±0.0 | 41.7±0.0 | 15.0±0.1 | 41.2±0.0 | 14.9±0.1 | **34.9**±0.0 | 14.7±0.1 |
| LSUN | 71.7 | 18.7 | 66.0±0.0 | 19.0±0.0 | 56.0 | 10.0 | 72.2±0.0 | 20.1±0.2 | 42.9±0.1 | 16.2±0.5 | 42.0±0.1 | 15.5±0.2 | **32.3**±0.1 | 11.9±0.3 |
| CIFAR100 | 71.3 | 20.4 | 65.7±0.0 | 20.1±0.0 | 57.6 | **12.6** | 71.8±0.0 | 22.2±0.0 | 43.2±0.0 | 17.7±0.1 | 42.5±0.0 | 17.5±0.1 | **35.2**±0.0 | 16.0±0.1 |
| FMNIST3D | 72.5 | 21.9 | 66.9±0.0 | 21.7±0.0 | 61.9 | 20.0 | 72.8±0.0 | 22.9±0.0 | 45.3±0.0 | 21.5±0.1 | 38.9±0.0 | 12.6±0.1 | **16.8**±0.0 | **0.0**±0.0 |
| UniformNoise | 68.9 | 14.0 | 62.7±0.1 | 13.6±0.2 | 48.1 | **3.8** | 68.8±0.1 | 14.9±0.2 | 41.0±0.1 | 12.5±0.5 | 39.5±0.1 | 11.4±0.4 | **27.3**±0.1 | 4.1±0.2 |
| **CIFAR100** | 81.3 | - | 78.9±0.0 | - | 80.2 | - | 82.2±0.0 | - | 74.4±0.0 | - | 73.4±0.0 | - | 62.8±0.0 | - |
| SVHN | 53.5 | 78.9 | 49.1±0.0 | 78.3±0.0 | 44.7 | **65.5** | 46.8±0.0 | 68.2±0.0 | 42.6±0.0 | 77.4±0.2 | 42.0±0.0 | 78.2±0.3 | **34.9**±0.0 | 79.7±0.2 |
| LSUN | 50.7 | 74.7 | 46.6±0.0 | 75.0±0.0 | 47.1 | 76.0 | 53.6±0.0 | 76.8±0.1 | 39.6±0.1 | **73.5**±0.5 | 38.0±0.1 | **73.7**±0.3 | **30.3**±0.0 | 75.7±0.6 |
| CIFAR10 | 53.3 | 78.3 | 49.3±0.0 | 78.0±0.0 | 51.3 | **76.9** | 56.0±0.0 | 78.8±0.0 | 44.1±0.0 | 77.9±0.2 | 43.0±0.0 | 78.3±0.3 | **34.9**±0.0 | 79.1±0.2 |
| FMNIST3D | 38.9 | 60.8 | 34.8±0.0 | 60.0±0.0 | 38.1 | 59.6 | 44.3±0.0 | 65.5±0.1 | 30.0±0.0 | 58.6±0.2 | 29.0±0.0 | 58.6±0.3 | **16.8**±0.0 | **38.7**±0.3 |
| UniformNoise | 29.4 | 55.8 | 25.7±0.1 | 55.5±0.4 | 45.1 | 94.9 | 31.6±0.1 | 49.9±0.1 | 22.0±0.1 | 47.0±0.4 | 17.1±0.1 | **24.0**±0.8 | **14.3**±0.0 | 29.6±0.5 |

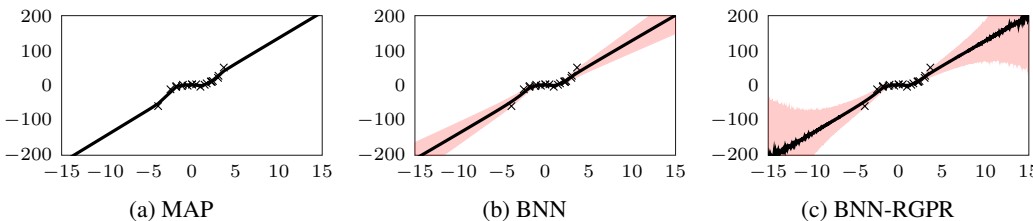

(a) MAP    (b) BNN    (c) BNN-RGPR

Figure 7: Toy regression with a BNN and additionally, our RGPR. Shades represent ±1 std. dev.

Table 8: OOD data detection results in terms of AUROC and AUPRC metrics. All values are averages and standard errors over 10 prediction trials.

| Datasets | MAP AUROC ↓ | MAP AUPRC ↓ | Temp. Scaling AUROC ↓ | Temp. Scaling AUPRC ↓ | Deep Ens. AUROC ↓ | Deep Ens. AUPRC ↓ | GP-DSCS AUROC ↓ | GP-DSCS AUPRC ↓ | LLL AUROC ↓ | LLL AUPRC ↓ | LLL-RGPR-LL AUROC ↓ | LLL-RGPR-LL AUPRC ↓ | LLL-RGPR-OOD AUROC ↓ | LLL-RGPR-OOD AUPRC ↓ |
|---|---|---|---|---|---|---|---|---|---|---|---|---|---|---|
| **MNIST** | - | - | - | - | - | - | - | - | - | - | - | - | - | - |
| EMNIST | 95.0 | 89.6 | 94.9±0.0 | 89.5±0.0 | **95.7** | **91.2** | 94.8±0.0 | 89.0±0.0 | 94.2±0.0 | 86.8±0.0 | 94.5±0.0 | 87.6±0.0 | 94.5±0.0 | 87.8±0.0 |
| KMNIST | 96.0 | 93.0 | 96.1±0.0 | 93.5±0.0 | 98.3 | 97.6 | 96.4±0.0 | 93.7±0.0 | 98.4±0.0 | 98.3±0.0 | **99.8**±0.0 | **99.8**±0.0 | **99.8**±0.0 | **99.8**±0.0 |
| FMNIST | 92.2 | 85.8 | 92.2±0.0 | 86.2±0.0 | 96.6 | 94.0 | 92.7±0.0 | 86.5±0.0 | 96.8±0.0 | 96.9±0.0 | 99.7±0.0 | 99.7±0.0 | **99.8**±0.0 | **99.8**±0.0 |
| GrayCIFAR10 | 98.0 | 98.5 | 97.8±0.0 | 98.4±0.0 | 99.0 | 99.4 | 98.0±0.0 | 98.6±0.0 | 98.5±0.0 | 99.0±0.0 | **99.9**±0.0 | **100.0**±0.0 | 99.8±0.0 | 99.9±0.0 |
| UniformNoise | 0.1 | 59.8 | 0.4±0.0 | 60.1±0.0 | 42.6 | 76.5 | 0.1±0.0 | 59.8±0.0 | 84.6±0.0 | 96.3±0.0 | **99.9**±0.0 | **100.0**±0.0 | 99.8±0.0 | **100.0**±0.0 |
| **CIFAR10** | - | - | - | - | - | - | - | - | - | - | - | - | - | - |
| SVHN | 95.7 | 91.0 | 96.1±0.0 | 91.2±0.0 | 95.2 | 92.0 | 93.6±0.0 | 85.6±0.0 | **96.3**±0.0 | **92.1**±0.0 | 96.2±0.0 | 91.9±0.0 | 95.8±0.0 | 90.2±0.0 |
| LSUN | 91.8 | 99.6 | 92.2±0.0 | 99.6±0.0 | **92.8** | **99.7** | 90.7±0.0 | 99.6±0.0 | 92.7±0.0 | **99.7**±0.0 | **92.8**±0.0 | **99.7**±0.0 | 92.6±0.0 | **99.7**±0.0 |
| CIFAR100 | 87.3 | 83.7 | 87.4±0.0 | 83.4±0.0 | **90.1** | 89.5 | 86.3±0.0 | 82.4±0.0 | 88.0±0.0 | 84.7±0.0 | 87.9±0.0 | 84.5±0.0 | 87.0±0.0 | 82.9±0.0 |
| FMNIST3D | 92.9 | 92.2 | 93.3±0.0 | 92.5±0.0 | 94.0 | 94.5 | 92.3±0.0 | 91.6±0.0 | 94.7±0.0 | 94.5±0.0 | 94.7±0.0 | 94.5±0.0 | **97.4**±0.0 | **97.5**±0.0 |
| UniformNoise | 96.7 | 99.2 | 97.1±0.0 | 99.3±0.0 | 92.8 | 98.4 | 94.2±0.0 | 98.7±0.0 | 98.8±0.0 | 99.7±0.0 | **98.9**±0.0 | 99.7±0.0 | **98.9**±0.0 | **99.8**±0.0 |
| **SVHN** | - | - | - | - | - | - | - | - | - | - | - | - | - | - |
| CIFAR10 | 95.4 | 97.0 | 95.4±0.0 | 96.9±0.0 | **97.5** | 98.9 | 95.0±0.0 | 96.7±0.0 | 97.3±0.0 | 98.9±0.0 | 97.3±0.0 | 98.9±0.0 | 97.4±0.0 | **99.0**±0.0 |
| LSUN | 95.6 | 99.9 | 95.6±0.0 | 99.9±0.0 | **98.0** | **100.0** | 95.1±0.0 | 99.9±0.0 | 97.4±0.0 | **100.0**±0.0 | 97.4±0.0 | **100.0**±0.0 | **98.0**±0.0 | **100.0**±0.0 |
| CIFAR100 | 94.5 | 96.4 | 94.5±0.0 | 96.4±0.0 | **97.3** | 98.7 | 94.1±0.0 | 96.1±0.0 | 96.8±0.0 | 98.7±0.0 | 96.9±0.0 | 98.7±0.0 | 97.1±0.0 | **98.8**±0.0 |
| FMNIST3D | 94.2 | 96.4 | 94.2±0.0 | 96.4±0.0 | 96.5 | 98.5 | 94.1±0.0 | 96.4±0.0 | 96.0±0.0 | 98.2±0.0 | 97.8±0.0 | 99.2±0.0 | **99.9**±0.0 | **100.0**±0.0 |
| UniformNoise | 96.8 | 99.7 | 96.9±0.1 | 99.7±0.0 | **98.9** | 99.9 | 96.7±0.1 | 99.7±0.0 | 97.7±0.0 | 99.8±0.0 | 97.9±0.0 | 99.8±0.0 | 98.8±0.0 | **99.9**±0.0 |
| **CIFAR100** | - | - | - | - | - | - | - | - | - | - | - | - | - | - |
| SVHN | 78.8 | 63.7 | 79.3±0.0 | 64.2±0.0 | **84.6** | 73.2 | 84.4±0.0 | **73.3**±0.0 | 80.3±0.0 | 66.6±0.0 | 79.9±0.0 | 65.7±0.0 | 78.0±0.0 | 58.7±0.0 |
| LSUN | 81.1 | 99.1 | 81.2±0.0 | 99.1±0.0 | **83.2** | **99.2** | 80.3±0.0 | 99.1±0.0 | 82.5±0.1 | **99.2**±0.0 | 82.9±0.1 | **99.2**±0.0 | 82.3±0.0 | **99.2**±0.0 |
| CIFAR10 | 78.7 | 77.8 | 78.9±0.0 | 77.9±0.0 | **80.1** | 79.6 | 78.1±0.0 | 77.2±0.0 | 78.9±0.0 | 77.6±0.0 | 78.9±0.0 | 77.7±0.0 | 77.9±0.0 | 75.6±0.0 |
| FMNIST3D | 87.4 | 86.9 | 87.8±0.0 | 87.3±0.0 | 89.0 | 89.5 | 85.7±0.0 | 85.4±0.0 | 88.5±0.0 | 88.1±0.0 | 88.6±0.0 | 88.2±0.0 | **93.3**±0.0 | **93.1**±0.0 |
| UniformNoise | 93.4 | 98.5 | 93.5±0.0 | 98.5±0.0 | 86.4 | 96.9 | 93.3±0.0 | 98.5±0.0 | 94.2±0.0 | 98.7±0.0 | **96.3**±0.0 | **99.2**±0.0 | 95.8±0.0 | 99.1±0.0 |

Table 9: Optimal hyperparameter for each layer (or residual block for ResNet ) on LLL.

| Datasets | Input | Layer 1 | Layer 2 | Layer 3 | Layer 4 |
|---|---|---|---|---|---|
| $\mathcal{L}_{\mathbf{LL}}$ | | | | | |
| MNIST | 3.3939e-08 | 5.4485e-07 | 1.1377e-07 | 2.3509e-03 | - |
| SVHN | 9.3995e-04 | 1.3767e-04 | 1.1347e-04 | 2.2835e-04 | 3.9480e-05 |
| CIFAR10 | 0.0036 | 0.0005 | 0.0008 | 0.0018 | 0.0028 |
| CIFAR100 | 0.0094 | 0.0093 | 0.0019 | 0.0049 | 0.0144 |
| $\mathcal{L}_{\mathbf{OOD}}$ (Synthetic) | | | | | |
| MNIST | 1.7384e-05 | 1.6409e-06 | 1.3555e-07 | 2.5206e-03 | - |
| SVHN | 8.2850e+00 | 6.2021e-03 | 9.1418e-03 | 4.7633e-03 | 1.3424e-02 |
| CIFAR10 | 4.6957e+01 | 8.4602e-04 | 1.3050e-03 | 5.9322e-03 | 1.9222e-03 |
| CIFAR100 | 2.6372e+01 | 2.8527e-03 | 8.7588e-04 | 4.5595e-03 | 2.5490e-01 |
| $\mathcal{L}_{\mathbf{OOD}}$ (32x32 ImageNet) | | | | | |
| MNIST | 3.5457e-08 | 5.9255e-07 | 1.1685e-07 | 2.4544e-03 | - |
| SVHN | 1.1849e-03 | 1.3038e-01 | 3.5909e-04 | 3.8309e-04 | 8.2367e-05 |
| CIFAR10 | 0.0236 | 0.9079 | 0.0030 | 0.0049 | 0.0053 |
| CIFAR100 | 0.0152 | 0.9533 | 0.0051 | 0.0094 | 0.2049 |

Table 10: UQ performance with ImageNet32x32 as $\mathcal{D}_{\mathrm{out}}$.

| Methods | MNIST | CIFAR10 | SVHN | CIFAR100 |
|---|---|---|---|---|
| **ECE ↓** | | | | |
| LLL-RGPR-LL | 15.8 | 3.6 | 13.1 | 5.7 |
| LLL-RGPR-OOD | 19.6 | 12.5 | 15.9 | 15.8 |
| LLL-RGPR-OOD ImageNet | 15.8 | 20.3 | 18.8 | 19.3 |
| **FPR@95 ↓** | | | | |
| LLL-RGPR-LL | 3.9 | 29.6 | 13.8 | 65.8 |
| LLL-RGPR-OOD | 3.6 | 24.2 | 9.6 | 63.0 |
| LLL-RGPR-OOD ImageNet | 3.9 | 39.5 | 7.3 | 61.0 |

Table 11: Regression far-away outlier detection. Values correspond to predictive error bars (averaged over ten prediction trials), similar to what shades represent in Fig. 2. "In" and "Out" correspond to inliers and outliers, respectively.

| Methods | housing In ↓ | housing Out ↑ | concrete In ↓ | concrete Out ↑ | energy In ↓ | energy Out ↑ | wine In ↓ | wine Out ↑ |
|---|---|---|---|---|---|---|---|---|
| LLL | 0.405 | 823.215 | 0.324 | 580.616 | 0.252 | 319.890 | 0.126 | 24.176 |
| LLL-RGPR | 0.407 | **2504.325** | 0.329 | **3394.466** | 0.253 | **2138.909** | 0.129 | **1948.813** |
| KFL | 1.171 | 2996.606 | 1.281 | 2518.338 | 0.651 | 1486.748 | 0.291 | 475.141 |
| KFL-RGPR | 1.165 | **3909.140** | 1.264 | **4258.177** | 0.656 | **2681.780** | 0.292 | **2031.481** |
| SWAG | 0.181 | 440.085 | 1.192 | 2770.455 | 0.418 | 1066.044 | 0.181 | 77.357 |
| SWAG-RGPR | 0.186 | **2403.366** | 1.146 | **4693.273** | 0.428 | **2647.922** | 0.187 | **1947.677** |
| SVGP | 0.641 | 2.547 | 0.845 | 3.100 | 0.367 | 2.237 | 0.092 | 0.983 |
| SVGP-RGPR | 0.641 | **1973.506** | 0.845 | **1932.061** | 0.367 | **1931.299** | 0.095 | **1956.027** |