# OpenReview forum: "An Infinite-Feature Extension for Bayesian ReLU Nets That Fixes Their Asymptotic Overconfidence"
_NeurIPS.cc/2021/Conference — NeurIPS 2021 Spotlight_

### Official Review · Reviewer_YyKC · 2021-07-12

**Rating:** 6
**Confidence:** 3

**Summary:**

The paper analyzes the problem of ReLU networks asymptotic overconfidence by formulating a model where the variance scales cubically w.r.t. distance from the training data which can be applied to existing models without requiring retraining.

**Limitations And Societal Impact:**

The limitations are discussed although I think they can be expanded as indicated in the main review. I am not aware of any potential negative societal impacts.

**Main Review:**

## Pros

- The authors explore a novel approach that is very useful and intuitive.
- The approach is theoretically well grounded.
-  The paper is very readable, and the authors included a thorough appendix with code

## Cons

- L19-20 "the confidence of BNN's can be bounded away from one" I found this sentence to be confusing and I think it should be reworded for clarity.
- L23-24 This sentence alone is confusing before reading further into the paper. If the variance is quadratic and the mean is linear, then shouldn't the variance be the dominant term? The next line says that the confidence from the mean outweighs the growing variance
- The method works on the premise of increasing variance of predictions in order to achieve a uniform distribution. This makes the method inapplicable to regression problems. I think this should be mentioned in the limitations.
- The authors state that reference [3] cannot be applied to BNN's, but why is this the case?
- The authors cite [1], and [2], but they do not compare with these baselines which are very relevant. Both [1], and [2] use a network trained with OOD inputs which I believe is very analogous to the CV done with OOD data in the paper. It would be nice to see these added as baselines for comparison. For example, depending on the performance of the other methods, it may influence a practitioners decision on whether to use RGPR on a newly trained model or to go with one of the aforementioned methods.
- The results for Deep Ensembles in the baselines seem to be a little bit weak. A result from [4] showed that DE basically outperforms everything else which is a recurring result from other papers as well. However, in figure 6 it appears that a last layer Laplace approximation outperforms DE on almost every metric on CIFAR10-C. How can this be true?

I am tending to vote for acceptance, but I think there are a few issues which need to be addressed in my comments above, especially the last two bullet points.

[1] Hendrycks, D., Mazeika, M., & Dietterich, T. (2018). Deep anomaly detection with outlier exposure. arXiv preprint arXiv:1812.04606.

[2] Hein, M., Andriushchenko, M., & Bitterwolf, J. (2019). Why relu networks yield high-confidence predictions far away from the training data and how to mitigate the problem. In Proceedings of the IEEE/CVF Conference on Computer Vision and Pattern Recognition (pp. 41-50).

[3] Meinke, A., & Hein, M. (2019). Towards neural networks that provably know when they don't know. arXiv preprint arXiv:1909.12180.

[4] Ovadia, Y., Fertig, E., Ren, J., Nado, Z., Sculley, D., Nowozin, S., ... & Snoek, J. (2019). Can you trust your model's uncertainty? Evaluating predictive uncertainty under dataset shift. arXiv preprint arXiv:1906.02530.


**Time Spent Reviewing:**

4-5 hours

---

> ### Author Response · Authors · 2021-08-10
> **Response to Reviewer YyKC**
>
> Thank you very much for your review. We will address minor concerns directly in the paper.  Here we will address questions, major concerns, and misunderstandings. We hope that the following points sufficiently address your concerns and convince you to raise your score. If not, please do let us know in reply and we’d be glad to discuss.
>
> **Linear mean, quadratic variance:** Note that in the probit approximation, the mean is scaled by the _square root_ of the variance (quadratically growing variance => linearly growing standard deviation). Hence we get a ratio of two linearly growing functions, and the confidence will only converge to a constant $\in (0, 1)$. We will make this clearer in the introduction.
>
> **RGPR for regression:** RGPR can naturally be applied to regression problems. In this case, it will endow BNNs with faster predictive uncertainty growth outside the data region. In fact, you can find a discussion of this topic in Appendix D.4.
>
> **Why can’t we apply [3] to BNNs:** First, we would like to clear up a misunderstanding: We do not claim that [3] cannot be applied to BNNs---we only mention that [3] achieved uniform asymptotic confidence, but they did not show that their results hold for BNNs. Thus, to our best knowledge, RGPR is the first method to guarantee uniform asymptotic confidence specifically for BNNs. Nevertheless, applying [3] in the context of BNNs is indeed not as straightforward as RGPR since (i) it is not a _post-hoc_ method, (ii) requires OOD data in the likelihood (and not just for hyperparameter tuning), and (iii) it uses generative models that need to be trained jointly with the network.
>
> **[1] and [2] as baselines:** Since RGPR is simply a _post-hoc_ enhancement to (Gaussian-based) BNNs, we focus on comparing RGPR-augmented BNNs with standard Bayesian baselines like DE, based on [4]. Comparison of RGPR-BNNs against [1] and [2] would be unfair since [1, 2] are (i) not _post-hoc_, (ii) use OOD objective to train _all_ the network’s weights---in contrast, the tuning of RGPR only amounts to just optimizing much fewer number of scalar hyperparameters ($L$ many). Nevertheless, RGPR is complementary to [1] and [2]: We can use them instead of standard MAP-trained networks as the base upon which Laplace approximations are applied, see Appendix D.6 of [5]. Then, RGPR can be applied on top of them, giving them guarantees regarding asymptotic confidence.
>
> **Last-layer Laplace (LLL) is better than DE?** Thanks for the question. Actually, the reported results, unfortunately, include a further “color inversion” on top of the corruptions which explains the low accuracy in Fig. 6 compared to Ovadia et al. When just doing the CIFAR-10-C corruptions, we get an avg. accuracy of around 74%---in line with recent work such as [1, Tab. 1]. Please find below, the updated results of the mean over all 5 intensity levels which confirm the qualitative picture given in Figure 6 in the paper.
>
> |              |     NLL ↓ |     ECE ↓ |   Brier ↓ | Confidence ↓ | Accuracy ↑ |
> |--------------|----------:|----------:|----------:|-------------:|-----------:|
> | MAP          |     1.066 |     0.226 |     0.402 |        0.887 |      0.739 |
> | Temp.        |     0.914 |     0.147 |     0.378 |        0.842 |      0.739 |
> | DE           |     0.909 |     0.110 | **0.354** |        0.840 |  **0.752** |
> | GP-DSCS      |     1.096 |     0.232 |     0.413 |        0.888 |      0.734 |
> | LLL          |     0.872 |     0.080 |     0.363 |        0.800 |      0.739 |
> | LLL-RGPR-LL  |     0.870 | **0.079** |     0.363 |        0.796 |      0.738 |
> | LLL-RGPR-OOD | **0.869** |     0.095 |     0.363 |    **0.717** |      0.738 |
>
> We see that LLL is indeed competitive to DE. This observation has also been independently observed by [6, 7]: well-tuned Laplace approximations can be competitive to DE in dataset-shift and/or OOD detection tasks. Since LLL itself has been shown by [5] to be competitive to other flavors of Laplace approximations, it is then to be expected that LLL is competitive to DE.
>
>
> **References**
>
> [5] Kristiadi, Hein, and Hennig. "Being Bayesian, even just a bit, fixes overconfidence in ReLU networks." ICML, 2020.
>
> [6] Lee, Humt, Feng, and Triebel. "Estimating model uncertainty of neural networks in sparse information form." ICML, 2020.
>
> [7] Daxberger, Kristiadi, Immer, Eschenhagen, Bauer, and Hennig. "Laplace Redux--Effortless Bayesian Deep Learning." arXiv preprint arXiv:2106.14806 (2021).

---

> > ### Comment · Reviewer_YyKC · 2021-08-16
> > **Thanks for your response**
> >
> > I find it a bit concerning that the text states using the CIFAR10-C dataset, when actually there is another color corruption added which gives the method better results than the plain CIFAR10-C dataset results above. I scanned through the text again and couldn't find anywhere that the extra color corruption was mentioned.
> >
> > - Can you point to a place in the text where this is discussed?
> > - Why was the color inversion added in addition to the CIFAR10-C datasets existing corruptions?

---

> > > ### Author Response · Authors · 2021-08-16
> > > **Further response**
> > >
> > > Thank you for your additional question and sorry for the confusion.
> > >
> > > The “color inversion” is an accidental corruption due to a bug in our CIFAR-10-C data loader. It is caused by line 523 in the`util/dataloaders.py` code that we included in our supplementary materials:
> > >
> > > ```
> > > self.data = np.uint8(self.data*255)
> > > ```
> > >
> > > That line should have been omitted since actually the raw images are already represented in `[0, 255]` range---because of this line, when they are viewed via `plt.imshow`, their colors look inverted.
> > >
> > > Note that this is an isolated bug---our rotated-MNIST and OOD detection results are not affected. Moreover, as shown by the table in our previous response, we still observe that RGPR can improve the base method in this setting. So, no changes to the text are required beyond updating the bottom half of Fig. 6.

---

> > > > ### Comment · Reviewer_YyKC · 2021-08-17
> > > > **Thanks for the clarification of results**
> > > >
> > > > As this experiment was isolated, and the new numbers seem more in line with what I would expect from DE, I will raise my score. One small point though...
> > > >
> > > > Wouldn't the bug above cause a much larger corruption than just a color inversion? any value of magnitude 255 would have essentially been $255^2$, right?

---

> > > > > ### Author Response · Authors · 2021-08-17
> > > > > **Thank you**
> > > > >
> > > > > It is an actual color inversion (plus a constant $1$), due to NumPy's behavior when casting general signed integers to 8-bit unsigned ones:
> > > > >
> > > > > ```
> > > > > import numpy as np
> > > > >
> > > > > img = np.array([255, 253, 6, 1], dtype='uint8')
> > > > >
> > > > > # The bug; `(img*255).dtype` is 'int64'
> > > > > np.uint8(img*255)
> > > > > Out[3]: array([  1,   3, 250, 255], dtype=uint8)
> > > > >
> > > > > # Color inversion
> > > > > 255-img
> > > > > Out[4]: array([  0,   2, 249, 254])
> > > > > ```

---

### Official Review · Reviewer_d2CF · 2021-07-15

**Rating:** 7
**Confidence:** 4

**Summary:**

The paper proposes a new method to improve the overconfidence problem of BNN.
By leveraging the connection between infinite-wide neural networks and Gaussian processes, the Authors propose a methodology to add back the components of the uncertainty that would have eventually been lost by the finite-wide architectures.

**Limitations And Societal Impact:**

The limitations are properly address, while the social impact is not discussed (and in my opinion, it's ok for this kind of work).


**Main Review:**


The idea is clear and the overall quality of the paper is high.
The methodological contribution is substantial and its theoretical justification is grounded. The empirical evaluation, on the other hand, feels a bit underwhelming, especially in the presentation.
The Authors did indeed a more extensive campaign in the supplement (including some regression datasets with in/out data) which maybe could be explicitly referenced in the main paper.
Below some comments/questions:

- I liked the discussion on the extension to non-asymptotic regimes, as the general assumption that a BNN is well-trained could be a bit restrictive.
    It's not clear to me, though, whether Theorem 3 on uniform confidence does indeed apply to this case as well. Maybe a small discussion on this could helpful here.
- The main assumption for this method to work is the Gaussian posterior approximation.
From a practical point of view, did you consider comparing with other approximation, like variational methods, rather then just Laplace? Do you expect your method to behave differently?
- Still on approximation, what was the reason to use only last-layer approximation? Beside being computationally more challenging, I'm expecting RGPR to work even better with other Laplace approximations. I'm asking this because the improvements in Figure 6 (without considering the OOD case, see below), while being consistent, seems overall limited. Could this be due to this?
- The extension of learning the hyper-parameters via OOD data feels to me a bit unnecessary but I appreciate the Authors also discuss the limitations of such approach.


## Minor
- If possible, Proposition, Lemma and Theorem shouldn't be used with the same latex reference counter (Theorem 3 should be indeed Theorem 1).
- The presentation quality of the Supplement is a bit lower than the main paper, with some typos and mistakes.


**Time Spent Reviewing:**

8

---

> ### Author Response · Authors · 2021-08-10
> **Response to Reviewer d2CF**
>
> Thank you very much for your positive review! We will address minor concerns and suggestions directly in the text.
>
> **Whether Theorem 3 on uniform confidence does indeed apply to non-asymptotic regimes:** Building a non-asymptotic counterpart to Thm. 3 is difficult. Nevertheless, one can indeed give a guarantee about the upper bound of non-asymptotic confidence, simply because we know that the variance of $\tilde{f}(\alpha \bf{x})$ is at least as high as the variance induced by the DSCS kernel at $\alpha \bf{x}$. We will add a discussion in the text and an exact statement in the appendix.
>
> **RGPR with other Gaussian approximations?** RGPR is indeed compatible with other Gaussian-based BNNs. In Appendix D, we show that RGPR is also effective in mitigating asymptotic overconfidence of other (non-Laplace) methods (SWAG [2], SVDKL [3]).
>
> **Why only last-layer Laplace (LLL)? Improvements in Figure 6 without OOD tuning is limited?** We use it for simplicity and since it has been shown to be a strong baseline, competitive to all-layer Laplace and hence, other Bayesian methods [1]. The most important practical reason, though, is that LLL’s very low computational cost fits well with the equally low cost of our proposed extension. Without the OOD tuning objective, RGPR does indeed give lower improvements. However, this does not seem to be caused by the last-layer approximation, but might be because, optimizing only the log-likelihood ($\mathcal{L}_{\text{LL}}$), the DSCS kernel of RGPR likely does not assign sufficiently high variance around the data region. This is one explanation for why tuning RGPR with the OOD objective is beneficial for prediction outside the data region.
>
> Please let us know in reply if we can clarify things further.
>
> **References**
>
> [1] Kristiadi, Agustinus, et al. "Being Bayesian, even just a bit, fixes overconfidence in ReLU networks." ICML, 2020.
>
> [2] Maddox, Wesley J., et al. "A simple baseline for Bayesian uncertainty in deep learning." NeurIPS 2019.
>
> [3] Wilson, Andrew G., et al. "Stochastic variational deep kernel learning." NIPS 2016.

---

> > ### Comment · Reviewer_d2CF · 2021-08-24
> > **Keeping my score**
> >
> > First of all, thanks to the Authors for the response. After reading the reply to my questions and the discussion with other reviewers, I'm overall positive with this work and I'll keep my score (7) unchanged. The only (minor) remark that I still have is on the choice of the last-layer Laplace for the experimental section. For an updated version, I would encourage the Authors to have a discussion/numerical experiment on this in the main paper (Table 2 of appendix could be a good example to showcase).

---

### Official Review · Reviewer_oTxs · 2021-07-16

**Rating:** 8
**Confidence:** 4

**Summary:**

The paper studies the overconfidence in neural networks and Bayesian neural networks with ReLU activation functions, which are known to be overconfident away from the training data. They propose a model that uses a Gaussian process for the residual of the neural network’s predictions with a double-sided cubic spline kernel. Interestingly this kernel is the limit of infinitely many ReLU features, and it scales cubically with the norm of the input, which dominates both the mean (linear scaling) and variance (quadratic scaling) of the BNN. This results in points far from the training data having uniformly random predictions in the prior and after inference.

**Limitations And Societal Impact:**

Yes.

**Main Review:**

Originality: I believe the method is new. The derivation of the double-sided cubic spline kernel also gives an interesting reason for overconfidence in ReLU BNNs, and a principled way to address their miscalibration. I have not seen this derivation in other work.

Quality: The theoretical analysis is easy to follow and seems correct. The experimental results are also strong and quite extensive, they support the main claims in the paper and provide evidence that the method proposed can be successful in practical applications. The results in Fig. 6 are particularly promising for NLL, Brier score, and ECE, but I’m slightly confused why the accuracy is so low. Can these results be split up over corruption intensity as in Ovadia et al.?

Clarity: The paper is well written and the technical details are presented well, making the overall arguments easy to follow. The figures are also strong.

It seems necessary to guarantee that the training data are close to the origin, so that the GP can model the residuals as close to 0 there. How can this be guaranteed in practice? Are results sensitive to this?

Significance: I believe the paper would be a good fit for NeurIPS. I have some minor questions outlined above, but given that they can be addressed, I recommend that the paper be accepted.


**Time Spent Reviewing:**

6

---

> ### Author Response · Authors · 2021-08-10
> **Response to Reviewer oTxs**
>
> Many thanks for your positive review! We’re delighted you like the work. To your specific points:
>
> **Why the accuracy is so low in Fig. 6. Can these results be split up over corruption intensity as in Ovadia et al.?** Thanks for the question. Actually, the reported results, unfortunately, include a further “color inversion” on top of the corruptions which explains the low accuracy compared to Ovadia et al. When just doing the CIFAR-10-C corruptions, we get an avg. accuracy of around 74%---in line with recent work such as [1, Tab. 1]. While it is difficult to read it off from the plots, our mean accuracy seems to be quite a bit higher than that of Ovadia et al. In particular, our accuracy for the highest intensity (level 5) is around 60% whereas Ovadia et al. report around 40%. Please find below, the updated results of the mean over all 5 intensity levels which confirm the qualitative picture given in Figure 6 in the paper. We will add the detailed results, broken down according to corruption intensities, to the paper.
>
> |              |     NLL ↓ |     ECE ↓ |   Brier ↓ | Confidence ↓ | Accuracy ↑ |
> |--------------|----------:|----------:|----------:|-------------:|-----------:|
> | MAP          |     1.066 |     0.226 |     0.402 |        0.887 |      0.739 |
> | Temp.        |     0.914 |     0.147 |     0.378 |        0.842 |      0.739 |
> | DE           |     0.909 |     0.110 | **0.354** |        0.840 |  **0.752** |
> | GP-DSCS      |     1.096 |     0.232 |     0.413 |        0.888 |      0.734 |
> | LLL          |     0.872 |     0.080 |     0.363 |        0.800 |      0.739 |
> | LLL-RGPR-LL  |     0.870 | **0.079** |     0.363 |        0.796 |      0.738 |
> | LLL-RGPR-OOD | **0.869** |     0.095 |     0.363 |    **0.717** |      0.738 |
>
> **The data need to be close to the origin. How can this be guaranteed in practice? Are results sensitive to this?** We do standardization so that the training data is centered and concentrated at the origin (see Sec. 3.3). Moreover, the hyperparameter tuning further calibrates the uncertainty induced by the DSCS kernel (by varying its strength, $\sigma^2$---see Sec. 3.4). The results do not seem to be sensitive to this: Tab. 5, 6 in the appendix show for OOD detection that RGPR consistently improves the base BNN across different datasets.
>
> Please let us know in reply if we can clarify things further.
>
> **References**
>
> [1] Benz, Philipp, et al. "Revisiting batch normalization for improving corruption robustness." Proceedings of the IEEE/CVF Winter Conference on Applications of Computer Vision. 2021.

---

> > ### Comment · Reviewer_oTxs · 2021-08-16
> > **Follow up**
> >
> > I am satisfied with the authors' answers to my questions and would like to increase my score accordingly. However, I find the omission of a known bug on CIFAR10-C very concerning. This should be fixed before publication, even if that means rerunning experiments. Without this it is impossible to compare to other results on CIFAR10-C.

---

> > > ### Author Response · Authors · 2021-08-16
> > > **Thank you**
> > >
> > > Thanks a lot for your follow-up. We were only aware of this bug thanks to your review---we are very grateful for that. We have fixed this and re-run the experiment: the results are shown in the table in our previous response, and they are comparable to other CIFAR-10-C results. We will of course update Fig. 6 accordingly.

---

> > > > ### Comment · Reviewer_oTxs · 2021-08-16
> > > > **Increased score**
> > > >
> > > > Excellent! I'll increase my score now.

---

### Official Review · Reviewer_4z2v · 2021-07-16

**Rating:** 6
**Confidence:** 2

**Summary:**

**Prior work:** deep ReLU softmax networks are guaranteed to be absolutely confident (have $p_c = 1$ for some class $c$) infinitely far away from training data. Bayesian analogues allow for limiting $p_c < 1$, but not necessarily $p_c = 1/C$, i.e. minimally confident, which might be desirable.

**This work** suggests to replace trained BNNs with outputs $f$ with $f + \hat f$, where $\hat f$ is a centered Gaussian with a certain closed-form covariance function, with variance growing fast enough (cubic in the input's norm) to make the model maximally uncertain  ($p_c = 1 / C$ for all classes $c$ from $1$ to $C$) in the limit of infinite-norm inputs, while having a negligible impact on predictions near the training data. In more practical terms, the augmented model can improve OOD detection and produce better uncertainty estimates on corrupted data. However, this can also lead to worse calibration scores on OOD data that is still relatively close to the training data.

During inference, the cost of this augmentation is negligible. However, it introduces $L$ new hyper-parameters, where $L$ is the depth of the network, and these parameters need to be tuned via cross-validation.

**Limitations And Societal Impact:**

One limitation is listed above - I would prefer if authors discussed the computational burden of tuning hyper-parameters more, and perhaps also demonstrated that their OOD/corruption improvements come from specifically using the DSCS kernel rather than from having more hyper-parameters to tune.

**Main Review:**

## Originality

To my knowledge the paper is sufficiently original, although I am not well familiar with relevant literature.

## Quality

Decent. The authors clearly identify the problem, solve it theoretically, demonstrate results empirically, and show benefits on practical applications like OOD detection and uncertainty estimates on corrupted data.

One drawback, in my opinion, is that the authors' solution is not well-motivated. IIUC any additive mean-zero Gaussian with super-quadratic variance would work here. Why did the authors choose the DSCS kernel specifically? Did you consider other alternatives?

## Clarity

For someone unfamiliar with the domain (me), clarity could be much improved.

Firstly, I believe one has to be familiar with prior work https://arxiv.org/abs/2002.10118 to comfortably read this paper. Otherwise, a lot of discussion in the abstract and introduction about variance scaling and its impact on overconfidence only become clear once the reader reaches Equation (3).

Secondly, I don't see any benefits in presenting this work as extending BNNs with infinitely-many features instead of simply working with the proposed DSCS kernel. For example, it is clear from the presentation of the paper that adding a mean-zero, super-quadratic kernel solves the overconfidence issue, but it is not clear how adding an infinite number of randomly shifted ReLU units does. Further, these are not vanilla ReLU units, but generalized (offset) ReLUs. Finally, in deep BNN setting, the proposed extension doesn't really extend intermediary layers with an infinite number of units, but rather adds residual branches from intermediary layers to the outputs. Overall, I found this interpretation more confusing/forced than useful, and hope the authors could either better highlight the value that this intuition provides, or instead move such discussion to appendix to make the main paper more clear and concise.



## Significance

The paper appears to have practical significance based on Figure 6 / Table 1, but I'm not well familiar with SOTA to be confident.

One important aspect for practical applicability is the computational overhead of tuning the $L$ hyper-parameters. Table 7 demonstrates that optimal parameters are not easy to guess. How exactly were they chosen (e.g. how did you sample hyper-parameters for CV, and how many times?)? Am I correct that these need to be tuned separately for any combination of a dataset and pre-trained BNN? Roughly, how much does it cost to tune these parameters compared to doing inference alone?

Perhaps one way to demonstrate the utility of the proposed method is to replace the DSCS kernel with some other mean-zero kernels, also having a single hyper-parameter to tune, but undesirable scaling (<= quadratic), and confirm that under the same tuning budget they produce worse results.

Curious, would it make sense to only have $\sigma^2 > 0$ in the last (top) layer, and $0$ for all others? Intuitively, one shouldn't care about Euclidean norm in the raw input space, and similarly, I am not sure why put any weight on intermediary layers instead of only looking at the last layer. Perhaps this could be a reasonable and cheap alternative.


**Time Spent Reviewing:**

8

---

> ### Author Response · Authors · 2021-08-10
> **Response to Reviewer 4z2v**
>
> Thank you very much for your review. We will address minor concerns directly in the paper.  Here we will address questions, major concerns, and misunderstandings. We will of course make them clearer in the text.
>
> **The interpretation of RGPR as extending BNNs is more confusing/forced than useful?** Why the DSCS kernel specifically? You are right that, in principle, we could have suggested any kernel $k(x_1,x_2)$ with the property $k(x,x) = \Theta(x^q)$ with $q>2$. However, note that our construction is explicitly motivated for ReLU networks. So it is natural, albeit not necessary, to frame our construction as an extension of ReLU networks by adding an unbounded number of additional ReLU units. While you say this is confusing or forced, we believe that, quite on the contrary, it makes for a compelling philosophical motivation: Finite-width ReLU networks are arguably incomplete (and their overconfidence is a symptom of this). If we assume that there are more data “out there” in the input space, then the network should consider the presence of additional (also ReLU) features outside of the training domain---infinitely many such features. And as our work shows, doing so can be done in a tractable, _post-hoc_ manner. We do take your point that we can do a better job of explaining this, and will clarify this exposition in the paper.
>
> **Hyperparameter tuning?** It is done via optimization (with random initialization): We use Adam to optimize the objective discussed in Sec. 3.4 for $50$ epochs. Note that this only needs to be done _once_ after training the BNN and we _do not_ tune it separately for each test set. For more detail please refer to the accompanying code (`eval_OOD.py`; `get_best_kernel_var` function). Finally, this process is cheap: It takes around two minutes on CIFAR-10 with ResNet-18.
>
> **Would it make sense to only have $\sigma^2 > 0$ in the last (top) layer, and $0$ for all others?** It appears that different representations of the input (and even the raw input itself) are beneficial for RGPR. Table 7 in the Appendix (D.3.2) shows this: after tuning, RGPR assigns non-negligible $\sigma^2$ to lower layers and the raw input.
>
> **OOD/corruption improvements come from specifically using the DSCS kernel rather than from having more hyper-parameters to tune?** The improvement is indeed due to the construction of the DSCS kernel and RGPR, along with the hyperparameter tuning scheme. By increasing $\sigma^2$ (a multiplicative factor to the kernel), the variance $k(x, x)$ grows faster, and thus the variance in the region near the data (i.e. around $0$) is higher. This leads to lower predictive confidence, implying better OOD detection (assuming that low variance at the data region is maintained, which is ensured by the objective in Sec. 3.4). Nevertheless, as discussed previously, we agree that other kernels with similar properties to the DSCS kernel can also yield similar improvements, albeit philosophically less natural than the DSCS kernel.
>
> Please let us know in reply if we can clarify things further.

---

> > ### Comment · Reviewer_4z2v · 2021-08-22
> > **Thank you for replies; still some questions on motivation / ablations**
> >
> > Thank you for your replies! Some of my questions are answered, but I still find the overall motivation not very clear / lacking ablations, which is why I am keeping my score for now (weak accept). Details below:
> >
> > > The interpretation of RGPR as extending BNNs is more confusing/forced than useful? ...
> >
> > Thank you, this does clarify your intuition a bit, but I still find it incomplete. Precisely, note that you are adding two features to your network:
> >
> > 1. You add ReLU units with kinks (at $c \neq 0$);
> > 2. You add infinitely many of them.
> >
> > To properly motivate this choice, you would need to show, theoretically and/or experimentally, that doing neither of these in separation is enough. Precisely:
> >
> > 1. Do finite Bayesian ReLU-networks with kinks suffer from the same overconfidence issue as regular ReLU-nets? (admittedly, can't be done post-hoc, but still interesting)
> > 2. Do infinite Bayesian ReLU-nets suffer from the same issues when ReLUs have no kinks? (the kernel from https://papers.nips.cc/paper/2009/hash/5751ec3e9a4feab575962e78e006250d-Abstract.html, equation 6; same kernel in https://arxiv.org/abs/1711.00165, section B)
> >
> >
> >
> > Further, I assume that my prior point
> > > Finally, in deep BNN setting, the proposed extension doesn't really extend intermediary layers with an infinite number of units, but rather adds residual branches from intermediary layers to the outputs.
> >
> > remains valid, so the intuition breaks for deep networks (if not, please correct me).
> >
> >
> >
> > > Hyperparameter tuning? It is done via optimization (with random initialization): We use Adam to optimize the objective discussed in Sec. 3.4 for  epochs. Note that this only needs to be done once after training the BNN and we do not tune it separately for each test set. For more detail please refer to the accompanying code (eval_OOD.py; get_best_kernel_var function). Finally, this process is cheap: It takes around two minutes on CIFAR-10 with ResNet-18.
> >
> > Thank you, it is great to know the process is cheap!
> >
> > > Would it make sense to only have $\sigma^2 > 0$ in the last (top) layer, and $0$ for all others? It appears that different representations of the input (and even the raw input itself) are beneficial for RGPR. Table 7 in the Appendix (D.3.2) shows this: after tuning, RGPR assigns non-negligible $\sigma^2$ to lower layers and the raw input.
> >
> > Thank you for clarifying. While the table does show that best settings are far from concentrating on a single non-zero values, I'm curious if you have a sense of whether these results are significantly better than if you were to tune a single non-zero $\sigma^2 > 0$ assigned to the top layer (or, for an ideal ablation, any other single layer including input)? E.g. it could be that results have multiple optima, and a setting with a single non-zero $\sigma^2$ would be one of them.
> >
> > The reason I'm asking this is because having many non-zero $\sigma^2$ is motivated only by variance growing too slowly around the data in section 3.3. But this growth can be tuned with a single $\sigma^2$ - why use multiple layers?
> >
> > > OOD/corruption improvements come from specifically using the DSCS kernel rather than from having more hyper-parameters to tune? ...
> >
> > Thank you, this partially answers my question:
> >
> > > Nevertheless, as discussed previously, we agree that other kernels with similar properties to the DSCS kernel can also yield similar improvements, albeit philosophically less natural than the DSCS kernel.
> >
> > As I discuss above, I still don't find this construction particularly natural, and for this reason I think this work would benefit from making this explicit in the paper, and ideally including comparison with some other alternative kernels with similar variance scaling.
> >
> > >  The improvement is indeed due to the construction of the DSCS kernel and RGPR, along with the hyperparameter tuning scheme.
> >
> > I think one part of my question remains unanswered - if you were to use a different kernel, one that would still be overconfident at infinity, but also having $L$ hyper-parameters to tune on the same objective, would it perform worse than the DSCS? The goal here is to eliminate the possibility that overconfidence at infinity is merely a theoretical issue, while in practical tasks the benefit rather comes from the more flexible nature of the kernel with tunable hyperparameters.

---

> > > ### Author Response · Authors · 2021-08-24
> > > **Thanks very much for your follow-up!**
> > >
> > > Thanks a lot for your follow-up comments and questions. We will add clarification, discussion, and ablation experiments as you suggested.
> > >
> > > **Do finite Bayesian ReLU-networks with kinks suffer from the same overconfidence issue as regular ReLU-nets?** Standard (Bayesian) ReLU nets do implicitly use ReLU features with non-zero kinks in general, due to the bias parameters. To see this, consider the $l$-th hidden ReLU units:
> > >
> > > $$
> > >     \mathbf{h}^{(l)} = \max(0, \mathbf{W}^{(l)} \mathbf{h}^{(l-1)} + \mathbf{b}^{(l)}) ,
> > > $$
> > >
> > > where $\max$ denotes the component-wise maximum. The r.h.s. above is indeed a collection of “generalized ReLU” features that we defined in Sec. 2.3, with kinks at $-b^{(l)}\_1, \dots, -b^{(l)}\_{N_l}$. Since standard (Bayesian) ReLU nets are overconfident, it follows that the answer to your question is affirmative.
> > >
> > > **Do infinite Bayesian ReLU-nets suffer from the same issues when ReLUs have no kinks? (the kernel from Cho and Saul, 2009, eq. (6))** Yes, this kernel will still be overconfident since it has a quadratic variance, i.e., $k(\mathbf{x}, \mathbf{x}) = \Vert \mathbf{x} \Vert^2$. Scaling $\mathbf{x}$ with $\alpha > 0$ and plugging this variance into our eq. (3), we see that we have the same problem as [1]: both numerator and denominator are linear in $\alpha$ and hence only converges to a constant (not necessarily the uniform confident) as $\alpha \to \infty$.
> > >
> > > **In summary:** The combination of both (i) infinitely many ReLU features and (ii) arranging these features regularly by shifting the location of their kinks, does indeed required to achieve cubic variance growth (and thus uniform asymptotic confidence).
> > >
> > > Another way to see this is via the derivation of the cubic spline kernel in our Appendix A (with 1D input for simplicity). With finite $K$ (number of ReLU features), the variance of $f(x)$ in eq. (11) is only $O(x^2)$. But if we take $K \to \infty$, the sum becomes an integral and by solving it, we see that the variance becomes $O(x^3)$. This validates the needs of (i). For (ii), by not shifting the ReLU features, i.e. with $c_i = \mathrm{const}$ for all $i=1, \dots, K$, we will not arrive at the integral below line 489 and thus we will not achieve the desired cubic variance growth---the variance instead stays at $O(x^2)$ since the summand in eq. (11) is constant, and the factor $K$ from this sum cancels with the factor of $1/K$ in the prior variance. (Note that the factor $1/K$ is required for the Central Limit Theorem to apply, as also noted by [2, footnote 1].)
> > >
> > > ---
> > >
> > > **In deep BNN setting, the proposed extension doesn't really extend intermediary layers with an infinite number of units, but rather adds residual branches from intermediary layers to the outputs.** Indeed you are correct that RGPR does not extend the width of hidden layers in the literal sense. Instead, RGPR extends the ReLU BNN in the function space with another function that arises from infinitely many ReLU features.
> > >
> > > ---
> > >
> > > **I'm curious if you have a sense of whether these results are significantly better than if you were to tune a single non-zero assigned to the top (or any other) layer?** Please find some preliminary results for OOD detection below (avg. FPR95, lower is better). The setting is identical to the one in the paper (Tab. 1, with $\mathcal{L}_\text{OOD}$). We will add the full ablation in the appendix.
> > >
> > > |           | All Layers | Only Last layer |
> > > |-----------|-----------:|----------------:|
> > > | CIFAR-10  |       24.2 |            34.2 |
> > > | CIFAR-100 |       63.0 |            68.2 |
> > >
> > >
> > > **I still don't find this construction particularly natural, and for this reason I think this work would benefit from making this explicit in the paper, and ideally including comparison with some other alternative kernels with similar variance scaling.** We will add a discussion that the DSCS kernel is not the only kernel that can achieve uniform asymptotic confidence. In particular, in the related work section, we will discuss the second-order arc-cosine kernel [3, eq. (7)], which has $O(\Vert \mathbf{x} \Vert^4)$ variance. Please let us know if you have any other kernel in mind.
> > >
> > > **If you were to use a different kernel, one that would still be overconfident at infinity, but also having  hyper-parameters to tune on the same objective, would it perform worse than the DSCS?** It is not surprising that other kernels can also achieve good non-asymptotic performance: any sufficiently flexible kernel can model the uncertainty near the data well. However, other kernels either (a) have undesirable variance scaling or (b) do not have the ReLU interpretation. For instance, the first-order arc-cosine kernel, i.e. the ReLU kernel of Cho and Saul, doesn’t fix the asymptotic overconfidence of ReLU BNNs when used in RGPR, as we have established before. While one can use a higher-order version of the arc-cosine kernel to attain uniform asymptotic confidence, ultimately, it loses the ReLU interpretation and thus is not quite as natural as the DSCS kernel in the context of ReLU BNNs.
> > >
> > > Please find a preliminary result on OOD detection (identical setting to the previous table), comparing the DSCS kernel with the ReLU kernel of [3].
> > >
> > > |           | DSCS | Arc-Cosine ($n=1$) |
> > > |-----------|-----:|-------------------:|
> > > | CIFAR-10  | 24.2 |               26.2 |
> > > | CIFAR-100 | 63.0 |               62.5 |
> > >
> > > ---
> > >
> > > **References:**
> > >
> > > [1] Kristiadi, et al. "Being Bayesian, even just a bit, fixes overconfidence in ReLU networks." ICML 2020.
> > >
> > > [2] Lee, et al. "Deep neural networks as Gaussian processes." ICLR 2018.
> > >
> > > [3] Cho and Saul. "Kernel methods for deep learning." NIPS 2009.

---

> > > > ### Comment · Reviewer_4z2v · 2021-09-03
> > > > **Thank you for further details!**
> > > >
> > > > Thank you for clarifications and further details - I look forward to seeing this changes in the next revision!

---

### Decision · Program_Chairs · 2021-09-28

**Decision:**

Accept (Spotlight)

**Comment:**

The proposed method attempts to fix the asymptotic overconfidence issue in ReLU networks. All reviewers agree that the paper is of high quality, theoretically interesting, and practically sound, and thus should be accepted. The remaining concerns are about motivation and experiments with other methods in addition to last-layer Laplace --- the authors are strongly encouraged to improve these points in the next iteration.

**Consistency Experiment:**

NeurIPS has a long history of experimentation. In 2014, NeurIPS ran an experiment in which 10% of submissions were reviewed by two independent committees to quantify the randomness in the review process. This year, we repeated a variant of this experiment to see how the quality of the review process has changed over time.  This paper was part of the experiment and was therefore assigned to two committees (consisting of reviewers, an Area Chair, and a Senior Area Chair) that reached independent decisions.  If both committees made the same recommendation, this recommendation was followed. If a single committee recommended acceptance, the paper was accepted (with the exception of a few cases in which the other committee identified what we considered a fatal flaw, e.g., an error in a key result).

This copy’s committee reached the following decision: **Accept (Poster)**

The other committee assigned to the paper recommended **Accept (Spotlight)**.  You can find the other set of reviews, along with any follow up discussion with the authors here:
https://openreview.net/forum?id=ehzq1YQrucI